

# Continuous ground monitoring of vegetation optical depth and water content with GPS signals

Vincent Humphrey[1,2], Christian Frankenberg[1,3]

*Correspondence to*: Vincent Humphrey (vincent.humphrey@geo.uzh.ch or
vincent.humphrey@bluewin.ch)

[1]Division of Geological and Planetary Sciences, California Institute of Technology, Pasadena
[2]Department of Geography, University of Zürich, Zürich, Switzerland
[3]Jet Propulsion Laboratory, California Institute of Technology, Pasadena, CA, United States

**Abstract.** Satellite microwave remote sensing techniques can be used to monitor vegetation optical depth (VOD), a metric which is directly linked to vegetation biomass and water content. However, these large-scale measurements are still difficult to reference against either rare or not directly comparable field observations. So far, in-situ estimates of biomass or water status often rely on infrequent and time-consuming samplings, which are not necessarily representative of the canopy scale. Here, we present a simple technique based on Global Navigation Satellite Systems (GNSS) with the potential to bridge this persisting scale gap. Because GNSS microwave signals are attenuated and scattered by vegetation and liquid water, placing a GNSS sensor under a vegetated canopy and measuring changes in signal quality over time can provide continuous information on VOD, and thus on vegetation biomass and water content. We test this technique at forested site in Southern California for a period of 8 months. We show that variations in GNSS signal to noise ratios reflect the overall distribution of biomass density in the canopy and can be monitored continuously. For the first time, we show that this technique can resolve diurnal variations in VOD and canopy water content at hourly to sub-hourly time steps. Using a model of canopy transmissivity to assess these diurnal signals, we find that temperature effects on the vegetation dielectric constant, and thus on VOD, may be non-negligible at the diurnal scale or during extreme events like heatwaves. The rainfall and dew deposition events also suggest that canopy water interception can be monitored with this approach. The technique presented here has the potential to resolve two important knowledge gaps, namely the lack of ground truth observations for satellite-based VOD, as well as the need of a reliable proxy to extrapolate isolated and labour-intensive in-situ measurements of biomass,





canopy water content, or leaf water potential. We provide recommendations for deploying such off-the-shelf and easy-to-use radar systems at existing ecohydrological monitoring networks such as FluxNet or

SapfluxNet.

## 1. Introduction

Complementary to observations in the visible and near-infrared spectrum, microwave-based remote sensing of the vegetation can be used to obtain direct information on aboveground biomass and vegetation water content (Konings et al., 2019;Frappart et al., 2020). Such information is essential to improve our

understanding of how ecosystems respond and adapt to both natural and anthropogenic changes, including for instance droughts, deforestation, or global warming. However, while satellite products can provide a global picture, their algorithms also need to be calibrated and evaluated against other reference measurements, thus raising the need for ground truth observations. In the case of vegetation optical depth (VOD), which is arguably one of the main microwave observables for vegetation currently, continuously

gathered ground truth data does not exist yet (Li et al., 2021). In this introduction, we provide a quick introduction to microwave observations, highlight current applications of VOD, and review recent attempts to compare satellite VOD against other data. We then present a ground-based technique relying on Global Navigation Satellite Systems (GNSS) with the objective to address the lack of ground-based VOD observations.


Microwave remote sensing methods are broadly categorized as either passive or active. Passive instruments (radiometers) measure the amount of microwave radiation that is naturally emitted by the Earth, whereas active instruments (radars) transmit a specific radio signal and measure the properties of the backscattered (reflected) signal. In both cases, the measured signals (brightness temperature or

backscatter) depend on various factors, but in particular on the emissivity/reflectivity of the surface, and the transmissivity ($\gamma$) of the vegetation, which acts as a layer of temporally changing opacity between the ground and the atmosphere. The transmissivity of vegetation is typically controlled by factors influencing either its dielectric constant (e.g. vegetation water content and temperature) or its structure (density, shape, size, and distribution of the vegetation elements in the canopy). The vegetation optical depth




(VOD) is a single parameter condensing all these different contributions and, in combination with the
     incidence angle ($\theta$), is used to express the canopy transmissivity as a function of the incidence angle:

$$\gamma = e^{\frac{-VOD}{\cos\theta}} \qquad\qquad (1)$$

This formulation of the transmissivity is the expression of a Beer-Lambert's law, where VOD represents
     an attenuation coefficient (specific to the observation wavelength) and the term $1/\cos\theta$ accounts for the
     path length through the canopy, as VOD (as defined here) only relates to the vertical path. Higher VOD
     values indicate that the canopy is less transparent to microwaves.

It is worth noting that this definition of VOD is mainly inherited from the perspective of microwave
     remote sensing algorithms, where VOD has to be estimated in order to obtain other variables of interest
     such as soil moisture (Jackson et al., 1982;Jackson and Schmugge, 1991;Owe et al., 2008). Because both
     field campaigns and theoretical considerations showed that in-situ estimates of VOD can be related to
     vegetation water content and biomass (Ulaby and Jedlicka, 1984;Schmugge and Jackson, 1992;Paloscia
and Pampaloni, 1992), this sparked interest in the development of more robust and long-term estimates
     of satellite-based VOD. Global maps of VOD have since become available from numerous satellite
     microwave sensors (Moesinger et al., 2020;Chaubell et al., 2020) but can hardly be validated, as
     systematic ground-based VOD observations do not exist at the moment. Only a few studies have managed
     to compare satellite-based VOD against in situ observations. Most notably, Tian et al. (2016) have found
a good agreement between satellite-based VOD and in situ measurements of green biomass in African
     Sahel. Instead, the majority of studies assessing or using satellite-based VOD have relied on comparisons
     with other remotely sensed variables (e.g. Grant et al., 2016) or model-data fusion products. For instance,
     Rodríguez-Fernández et al. (2018) have compared VOD from the SMOS satellite against optical
     vegetation indices, lidar tree height and different aboveground biomass benchmark maps. Consistent with
other studies , their results show that VOD is often a better proxy for tree height and biomass, compared
     to optical greenness indices (such as NDVI or EVI). Thus, VOD has been increasingly used to monitor



changes in aboveground biomass and terrestrial carbon sequestration at the inter-annual and seasonal time scales (Brandt et al., 2018;Wigneron et al., 2020;Fan et al., 2019).

In addition to providing information on long-term biomass changes, VOD has also been shown to exhibit significant short-term variability, which is thought to be related to changes in relative vegetation water content (Feldman et al., 2021). Using observations from the AMSR-E satellite, Konings and Gentine (2016) found significant variations between midnight and midday VOD which may put an invaluable constrain on plant response to water stress at the ecosystem-scale (Lee et al., 2013;Konings et al., 2021).

Further studies also highlighted intercepted water (due to either rainfall or dew) as a potential factor explaining diurnal variations in VOD (Xu et al., 2021). Using a ground-based radiometer, Holtzman et al. (2021) showed that VOD variations over the course of a day could be linearly related to in-situ measurements of leaf water potential. This is consistent with a previous study by Momen et al. (2017) which found good agreement between satellite-based VOD and leaf water potential measurements across

three different U.S. sites. Measurements with active microwave sensors have also shown that radar backscatter exhibits diurnal variations which can be related to both dew and relative water content (Vermunt et al., 2021;Konings et al., 2017b).

Considering some advantages of microwave-range compared to visible-range observations[1], such studies

have demonstrated the interest of VOD for monitoring vegetation dynamics from space (Konings et al., 2021). However, they have also revealed our limited ability to (1) validate space-borne VOD observations and (2) disentangle the multitude of factors which may affect them across a wide range of ecosystems and climatic conditions. Established eco-hydrological measurement networks (e.g. FluxNet or SapFluxNet) can provide most of the necessary collocated observations in terms of meteorological

parameters, water fluxes, canopy structure, and biomass (e.g. Momen et al., 2017), but only few of these

---

[1] Microwave remote sensing has two key interesting properties, first it is relatively independent of cloud cover and solar illumination conditions, second it is not only sensitive to the surface of the observed material but also to its content, up to a certain penetration depth. Drawbacks include a lower signal energy for passive microwave remote sensing, which usually translates into coarser spatial resolutions, and some difficulty in disentangling the many different factors contributing to the measured signal (i.e. the ground versus the vegetation contributions, the surface roughness, the material's temperature, and its moisture content).





sites have ever been equipped with microwave radiometers or active radars, and if so, usually for limited periods of time. Yet, continuous in-situ VOD observations at these sites could serve as a particularly useful proxy to interpolate and gap-fill the sparse and labour-intensive measurements of biomass and leaf water status. There is thus a need for a cheap and robust method to obtain ground-based VOD
measurements over a wide variety of already monitored sites.

Here, we propose to use microwave signals from existing Global Navigation Satellite Systems (GNSS) to monitor the transmissivity and VOD of a vegetation canopy. The experimental setup consists in a pair of GNSS receivers, one placed on a tripod below a forest canopy, and one located above the canopy with
an unobstructed view of the sky (Fig. 1). The main idea is that the difference in measured signal strength between the two instruments will yield information on the opacity of the canopy. Fortunately, many survey-grade GNSS receivers available on the market can be configured to log signal strengths, making it relatively easy to implement such a system. The GNSS microwave signals fall in the L-band (1-2 GHz), similar to frequencies used by the SMOS and SMAP satellites for calculating VOD. Nowadays, four
major GNSS constellations (GPS, GLONASS, Galileo, and BeiDou) represent about a hundred of orbiting satellites, such that about 20 to 40 satellites may be visible and individually tracked from the ground at any given time and from any location in the world. Set-ups similar to the one shown in Fig. 1 have been tested before, for instance Rodriguez-Alvarez et al. (2012) used it to estimate the canopy water content of a walnut tree stand and Camps et al. (2020) used it to estimate VOD in a beech forest and make
comparisons with optical indices. Over the last decades, GNSS-based monitoring of the Earth's surface has been demonstrated in a wide variety of domains ranging from oceanography to hydrology. While we rely here on the attenuation of the direct GNSS signal through the canopy, it is worth noting that other GNSS-based techniques, such as GNSS reflectometry, have been used to monitor soil moisture, snow height, vegetation water content, and biomass (Larson et al., 2009;Small et al., 2010;Chew et al.,
2014;Egido et al., 2014;Larson, 2016;Chew and Small, 2018;Ruf et al., 2018;Santi et al., 2019;Carreno-Luengo et al., 2020;Guerriero et al., 2020;Pan et al., 2020;Munoz-Martin et al., 2022). GNSS reflectometry relies on GNSS signals that are reflected from the Earth's surface and which are weaker than the direct GNSS signals analysed here.



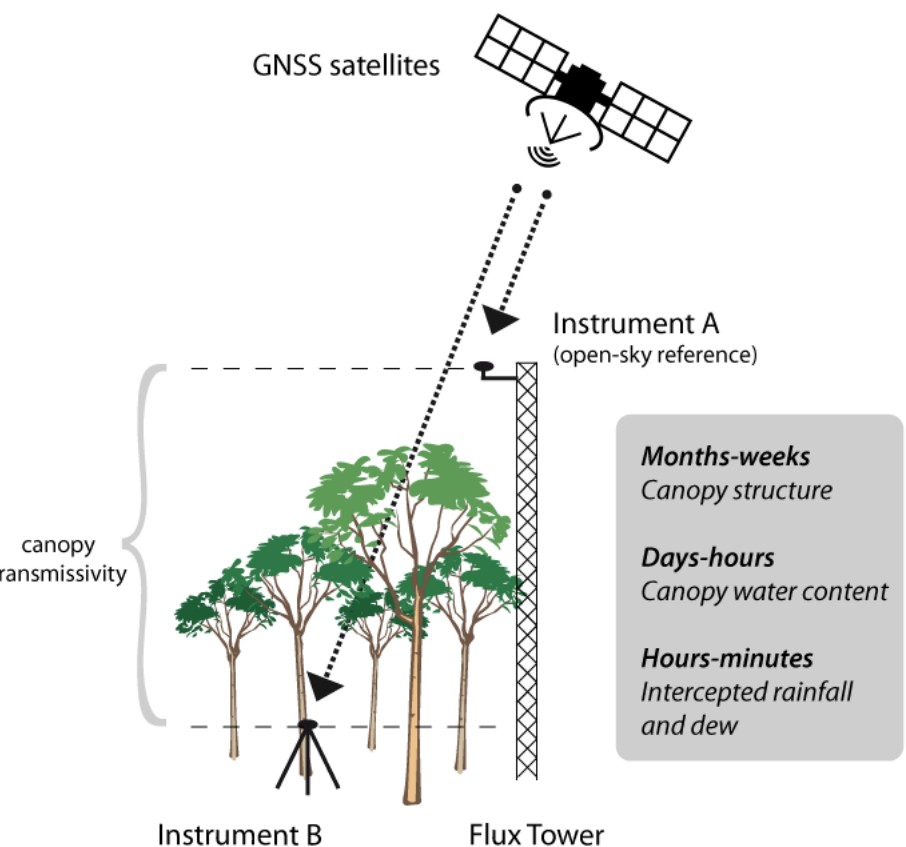

**Figure 1. Instrument setup for measuring GNSS-based VOD. Each instrument consists of an antenna and a GNSS receiver.**

The goal of this study is to demonstrate the potential of ground-based GNSS receivers for monitoring VOD, dry aboveground biomass, and water content continuously. In Section 2, we describe the measurements conducted at the study site and outline a simple canopy transmissivity model which is later

used to estimate canopy density and water content from VOD measurements. Section 3 presents the raw GNSS measurements and the processing approach used to transform these into a canopy-averaged VOD time series. Section 4 compares the obtained seasonal and diurnal VOD time series against other in-situ and satellite measurements. Section 5 provides an example of retrieval algorithm for aboveground biomass and canopy water content. Finally, section 6 summarizes the main conclusions and provides

some recommendations with respect to future deployments at existing ecohydrological sites.





## 2. Data and methods

### 2.1 Site set up

The experiment consists of a reference site which has an unobstructed view of the sky, and a forested site which is located under a semi-closed forest canopy (Fig. 1). The open-sky reference site is located on the
roof of a building of the California Institute of Technology in Pasadena, California (34.13624°, -118.12693°), and the forested site is located in the Huntington Library botanical garden, some 1.7 km away (34.12404° -118.11582°). The forested site is non-irrigated, with trees of about 5 to 15m height. Tree species surrounding the antenna are mainly live oaks (*quercus agrifolia* Née) and the understory is herbaceous. The overall climate is Mediterranean with weather conditions usually clear, daily maximum
temperatures between 25°C and 35°C, and low relative humidity. At each site a Septentrio PolaRx5e GNSS receiver, connected to a PolaNt-x MF GNSS antenna, measured multi-constellation GNSS signals over the period of May 13 to December 10, 2020, with a logging rate of 15 seconds. Power loss at the site from June 15[th] to July 1[st] caused a data gap of 17 days. The raw GNSS data was quality-checked using the 'teqc' pre-processing software, publicly available at the UNAVCO website
(www.unavco.org/software). The satellites' azimuth and elevation angles are also computed with teqc. Weather data is measured at the reference site by a station of the Total Carbon Column Observing Network (http://tccon-weather.caltech.edu/). Weather data acquisition was interrupted from July 15[th] to 26[th]. Rainfall was measured with a tipping-bucket at the DPW HQ station (5km from the forested site) by the Department of Public Works of the City of Los Angeles (https://dpw.lacounty.gov/wrd/rainfall/).

### 2.2 Leaf water content

Leaf samples were collected from two live oaks closest to the GNSS antenna on October 18, 2020, at 7am, 12pm and 5pm using a 2m long pruner. Leaves were weighed on-site immediately after being sampled (fresh weight; FW) and stored in cooled glass vials. They were then transported to the lab where about 1 cm of water was added to each vial to cover the leaves' petioles. Turgid weight (TW) was
measured after 12 hours. Then, the leaves were dried at 80°C for a period of 24 hours, after which dry weight (DW) was measured. Relative leaf water content (RLWC, in percent) is calculated as.




$$RLWC = \frac{FW - DW}{TW - DW} \cdot 100 \qquad (2)$$

We also calculate the gravimetric moisture content of the leaf ($m_g$, in g g⁻¹), a variable that is later used
to model the leaf dielectric constant.

$$m_g = \frac{FW - DW}{FW} \qquad (3)$$

### 2.3 Canopy transmissivity model

We use a physical model of the canopy transmissivity to investigate the potential roles of canopy

volumetric density, water content, and temperature on the GNSS-based VOD measurements. We use a
relatively simple formulation which only considers the attenuating effect of the canopy on the direct signal
power and represents the canopy as a homogeneous layer, assumed to consist of randomly distributed
elements (Ulaby and Jedlicka, 1984;Ulaby and Long, 2014;Guglielmetti et al., 2007). The transmissivity
of the canopy is expressed as a function of a bulk canopy extinction coefficient ($\kappa_e$), canopy height ($h$, in

meters), and the incidence angle.

$$\gamma = e^{\frac{-2\,\kappa_e\,h}{\cos\theta}} \qquad (4)$$

We define a fixed average canopy height of 7 meters based on field observations. Neglecting scattering,
the extinction coefficient is related to the complex index of refraction of the canopy layer ($n_c''$) (Ulaby
and Long, 2014).

190                    $$\kappa_e = \frac{2\pi}{\lambda_0} n_c'' \qquad (5)$$

where $\lambda_0$ is the free-space wavelength (in meters). Note that this formulation and the overall concept of
bulk coefficients is applicable only when the inclusions in the canopy (i.e. the pockets of water within the
vegetation tissues) are smaller in size compared to the observation wavelength (here $\lambda_0 \approx 19$ cm for the
GPS L1 frequency), so that scattering effects are small enough to be neglected (also see Jackson and

Schmugge, 1991;Ulaby and Long, 2014). Using a theoretical scattering model, Guerriero et al. (2020)
also confirmed that the right-hand circular polarized (RHCP) GNSS signals measured below a forest
canopy are dominated by coherent attenuation whereas only the left-hand circular polarized signals





(which GNSS antennas are designed to reject) are dominated by volume scattering. The complex index of refraction is calculated from the dielectric constant of the canopy layer ($\varepsilon_c$) (Ulaby and Long, 2014).

$$n_c'' = -\Im\{\sqrt{\varepsilon_c}\} \tag{6}$$

The canopy is constituted of two main phases; the surrounding air, which makes up most of the canopy volume, and the vegetation material. The dielectric of the canopy $\varepsilon_c$ is calculated using a two-phase refractive mixing approach (Ulaby and Long, 2014, Eq. 4.45).

$$\sqrt{\varepsilon_c} = v_{veg}\sqrt{\varepsilon_{veg}} + \left(1 - v_{veg}\right)\sqrt{\varepsilon_{air}} \tag{7}$$

where $v_{veg}$ represents the volume fraction of vegetation within the canopy (on the order of 0.0001-0.01 m³/m³) and is to be seen as a vegetation density parameter that may vary as a function of the growth cycle. The term $\left(1 - v_{veg}\right)\sqrt{\varepsilon_{air}}$ is practically equal to 1 with no imaginary part such that Eq. 7 can be rewritten as (Ulaby and Long, 2014, Eq. 11.89).

$$n_c'' \approx -\Im\{\sqrt{\varepsilon_{veg}}\} v_{veg} \tag{8}$$

The dielectric of the vegetation ($\varepsilon_{veg}$) incorporates a real and an imaginary part, namely the dielectric permittivity and the dielectric loss ($\varepsilon_{veg} = \varepsilon'_{veg} - i\varepsilon''_{veg}$). Both depend on various factors, but most importantly on the considered wavelength, the vegetation water content, as well as the plant water's temperature and ionic conductivity. Here we model $\varepsilon_{veg}$ using the semi-empirical model for vegetation introduced by Ulaby and El-rayes (1987) and valid over the range 0.2-20 GHz. This model is derived

from an observational dataset of corn leaves, and has been successfully used for a wide range of species, including trees (e.g. Chuah et al., 1995). The model expresses the dielectric of the vegetation $\varepsilon_{veg}$ as a function of its gravimetric moisture ($m_g$), its temperature, and its salinity. The numerous equations are not reported here but can be found for instance in Ulaby and El-rayes (1987), Ulaby and Long (2014), or Steele-Dunne et al. (2012).

The purpose of the transmissivity and dielectric models described above is to provide a first-order estimate of the potential effects of canopy density, temperature and water content changes on canopy transmissivity. In Fig. 2, we illustrate the modelled VOD response (at the GPS L1 1.575 GHz frequency) to potential changes in vegetation water content, temperature, and vegetation volumetric density. As expected, an increase in vegetation moisture content (Fig. 2a) leads to a substantial increase of the



vegetation's dielectric permittivity and dielectric loss, which results in a lower canopy transmissivity. Temperature influences the vegetation's dielectric properties as well, although less markedly (Fig. 2b). The ionic conductivity of the plant water is the main factor explaining the slight dependency of the loss factor (and of VOD) to the temperature. Finally, vegetation density within the canopy is also a parameter that strongly controls the transmissivity (Fig. 2c). Note that the shape of these response curves (the

response to temperature in particular) may change depending on the considered frequency.

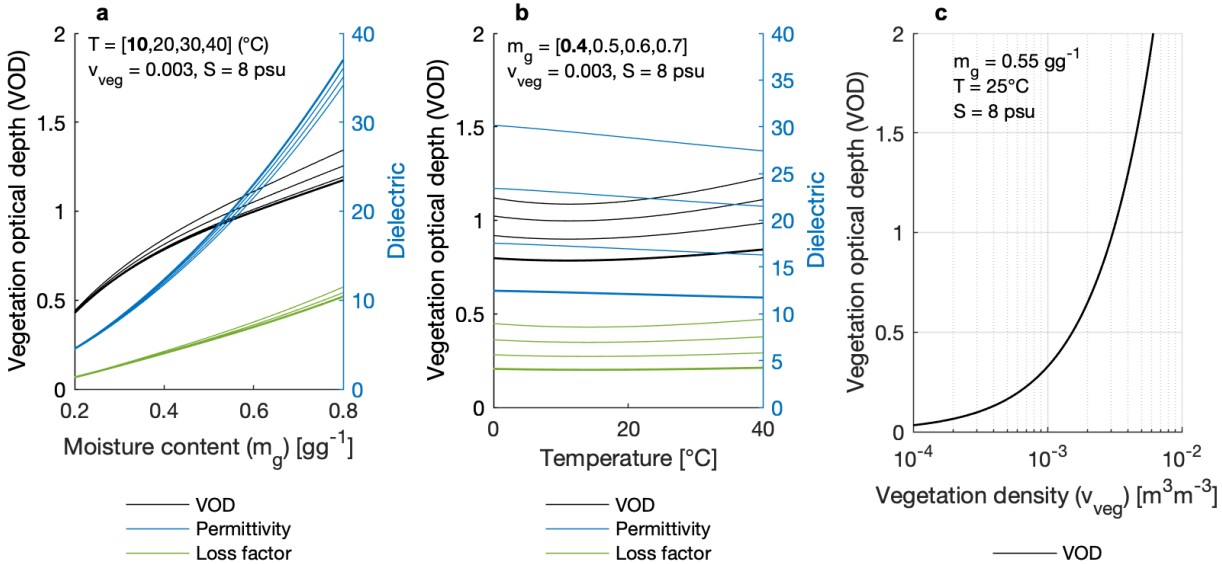

**Figure 2.** Canopy transmissivity at the GPS L1 frequency (1.575 GHz). a-b) modelled responses of canopy VOD and the vegetation material's dielectric properties to changes in gravimetric moisture content and temperature. The bold lines in (a) are obtained assuming a temperature of 10°C, with thin lines indicating 10°C increments. The bold lines in (b) are obtained assuming a

gravimetric moisture content of 0.4 $gg^{-1}$, with thin lines indicating 0.1 $gg^{-1}$ increments. c) modelled response of canopy VOD to canopy density. Canopy height is fixed at 7 meters and salinity to 8‰.

## 3. GNSS data processing

### 3.1 Raw SNR observations

Most survey-grade GNSS receivers commonly register signal-to-noise ratios (SNR, in decibel) which

express the magnitude of the received signal power from each satellite compared to the background noise (Bilich et al., 2007). The hemispherical plot in Fig. 3a illustrates the SNR values measured over the course of a single day at the reference (open-sky) station for one satellite of the GPS constellation. As is very commonly observed, the SNR increases as the satellite rises up above the horizon (12PM mark), reaches





its peak value at maximum elevation (2PM mark), where the antenna gain is the strongest, and decreases

again until the satellite disappears from view (5PM mark). As can be seen from Fig. 3b, the same satellite

track observed from the forested site shows numerous drops in SNR. Assuming a comparable level of

background noise at the two sites, the SNR difference between the two sites ($\Delta SNR$, Eq. 2) mainly reflects

the transmissivity of the canopy, expressed in decibels. As expected, it is mostly negative (Fig 1c-d),

indicating attenuation by the forest canopy.


$$\Delta SNR = SNR_{Forested} - SNR_{OpenSky} \qquad (9)$$

Combining all available data (May to December 2021) from 102 individual GNSS satellites, we produce

a hemispherical map of the average $\Delta SNR$ (Fig. 4a), which matches the overall distribution of canopy

density as seen from the antenna location (Fig. 4b).





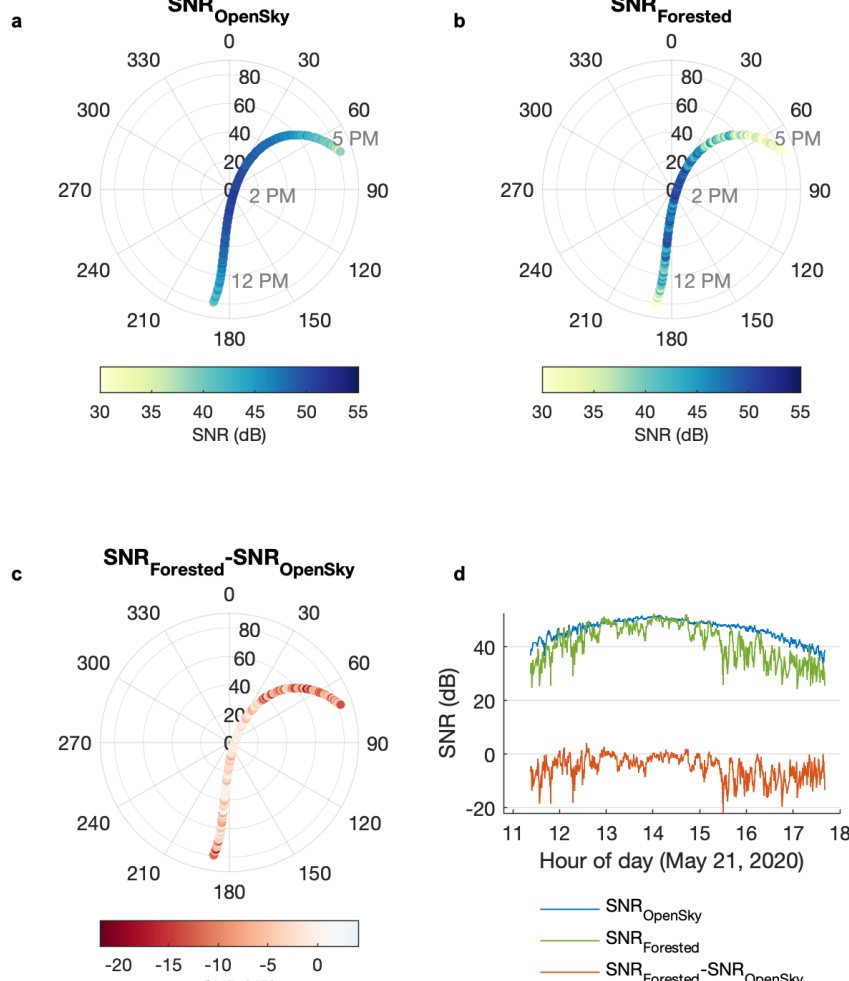

**Figure 3. Sky plots illustrating the SNR observations on May 21, 2020 for one specific GPS satellite (PRN2) at the open sky site (a) and the forested site (b). (c) Difference in SNR between the two sites. (d) Same as a-c but showing the temporal evolution of the SNR.**

It should be noted that while the SNR measurement is dominated by the contribution of the direct (line of sight) signal, it also includes a comparatively weaker contribution from indirect multipath reflections (Bilich et al., 2007;Smyrnaios et al., 2013), which may contain information on soil water status for instance (Larson, 2016). Multipath often manifests itself as a periodic oscillation of the SNR which is caused by the successively constructive and destructive interference from ground reflections, as a function

the satellite's elevation angle. Such periodic oscillations, of a few dB, are barely visible for instance in Fig. 3d at the beginning and the end of the SNR$_{OpenSky}$ time series (blue curve), where the roof's floor acts





as the reflector. However, while present in our data, multipath represents a signal that is about one order of magnitude smaller than the attenuations caused by the presence of trees in the line of sight. It is only in some favourable situations (i.e. flat grasslands, open water bodies, …), where large flat surfaces

surrounding the antenna produce a coherent structure in ground reflections, that multipath is strong enough to be reliably detected in SNR, even though GNSS systems are explicitly designed to reject such signals. Indeed, most geodetic-grade antennas have metal ground planes and are much less sensitive to the predominantly left hand circular polarized (LHCP) ground reflections of the transmitted right hand circular polarized (RHCP) signal. Thus, in our case, the difference in SNR between the two sites is

predominantly due to the attenuation of the direct RHCP signal by the forest canopy and it is reasonable to assume that multipath effects are of second order. This is also confirmed by a $\Delta SNR$ close to zero in the sky sectors where the canopy is either absent or very sparse (Fig. 4). It is only when the incidence angle is larger than 80° that the majority of the reflected GNSS signal is co-polarized (RHCP) (Smyrnaios et al., 2013). As a precaution, we discard all observations with an incidence angle higher than 80° for the

remainder of the analysis.

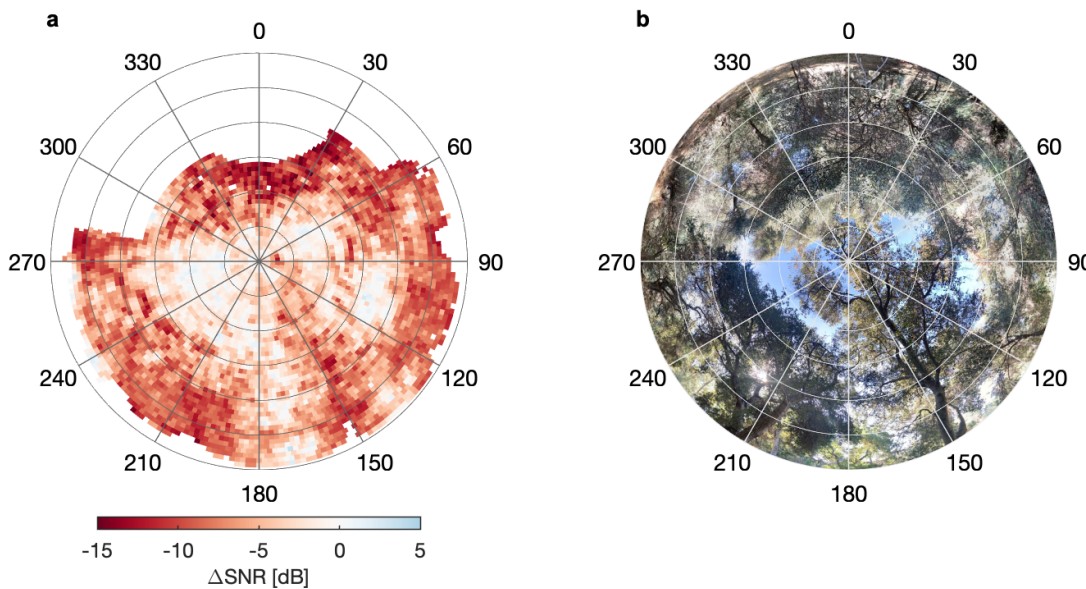

**Figure 4. (a) Sky plot illustrating the mean SNR difference between the open sky and the forested site for L1/L1C signals (n = 2.17·10[7] observations, taken over an 8-month period). The mean value within each 2-degree equal area sky sector is shown. Some sky sectors in (a) are obstructed by buildings at the reference site and are thus excluded from the analysis (towards the north-west and north-**

**east). (b) Hemispheric photograph taken from the perspective of the forested site antenna.**





### 3.2 Transmissivity and vegetation optical depth

After conversion from decibel to linear scale (Eq. 10), the $\Delta SNR$ measurements are used as the transmissivity estimate, and VOD is calculated from Eq. 11 (Eq. 1).

$$\gamma = 10^{\frac{\Delta SNR}{10}} \tag{10}$$


$$VOD = -\ln(\gamma)\cos\theta \tag{11}$$

The resulting hemispherical distribution of long-term mean transmissivity and VOD is reported in Fig. 5a-b. In some cases, the transmissivity values computed from the raw GNSS measurements were higher than 1, leading to a VOD lower than zero, which is unphysical (about 8% of all measurements). This occurs because individual SNR measurements unavoidably include some random noise as well as non-

random multipath interferences that can cause the measured signal power at the reference site to be transiently lower than at the forested site. This especially occurs where there are gaps in the canopy and both antennas have a clear line of sight to the satellite. To preserve the error structure of the measurements, we propose to still use these unphysical values whenever possible, and especially when computing averages, so that positive and negative random errors can cancel out, avoiding a potential bias in our

estimate of the long-term average VOD. While transmissivity has an obvious dependence on the incidence angle (Fig. 5c), this is not the case for VOD (Fig. 5d), as would be expected from Eq. 1. The strong anisotropy of the long-term VOD pattern (Fig. 5b) reflects the heterogeneous structure of the canopy (Fig. 4b), with local mean VOD values ranging from 0.16 to 2.46 (1[st] and 99[th] percentiles) depending on the azimuth and incidence angle. The whole canopy average VOD is 0.79, which is a credible value for

evergreen broadleaf forests (Konings et al., 2017a).





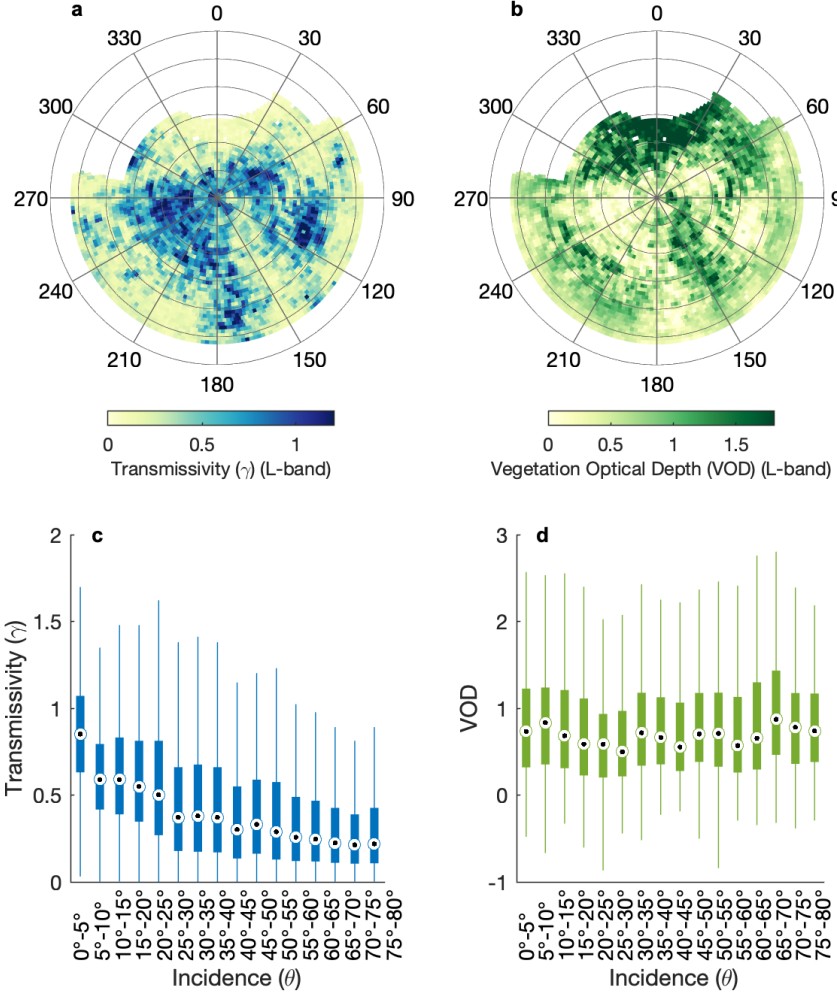

**Figure 5. Sky plot illustrating the mean canopy transmissivity (a) and mean VOD (b). The mean value within each 2-degree equal area sky sector is shown. (c-d) The box plots show the distribution (5th, 25th, 50th, 75th and 99th percentiles) of all individual transmissivity and VOD measurements (n = 2.17·10^7) as a function of the incidence angle (relative to zenith).**

### 3.3 Computation of anomaly time series

Because changes in VOD over time may provide valuable information on the vegetation's growth cycle and water content, it is of great interest to investigate its temporal evolution. However, this is complicated by the patterns of GNSS orbits, which change continuously. In Fig. 6a, we provide an overview of the most frequent orbit patterns over our site and their mean revisit time, showing which sky sectors are most often observed. While it takes about a day on average to cover most of the observable sky sectors (Fig. 6b, red curve), monitoring one specific section of the canopy every 1 or 2 hours would only be possible





over a narrow band where many satellite tracks coincide (Fig. 6a). Note that the location of the highly

sampled band (and the blind spot above it) depends on the site's latitude (it is closer to zenith at higher

latitudes) and would be located on the opposite side in the Southern hemisphere. As different cross-

sections within Fig. 5b are observed each day, the changing and irregular sampling of the canopy

introduces spurious variability in daily site-averaged VOD. When calculating sub-daily (e.g. hourly) time

series, this problem is even more important and will obfuscate most of the potential real variability. For

example, binning the raw GNSS-based VOD observations into hourly averages produces a rather noisy

time series with just seasonal trends visible (Fig. 7a-b, 'VOD raw'). This is because a lot of the variability

in 'VOD raw' is caused by the fact that different areas of an heterogenous canopy are observed every

hour. This issue can also be diagnosed quantitatively. For instance, computing the serial autocorrelation

of the raw VOD time series (Fig. 7c) reveals periodicities likely not related to ecohydrological processes

but instead caused by the combined repeat times of the different GNSS constellations.

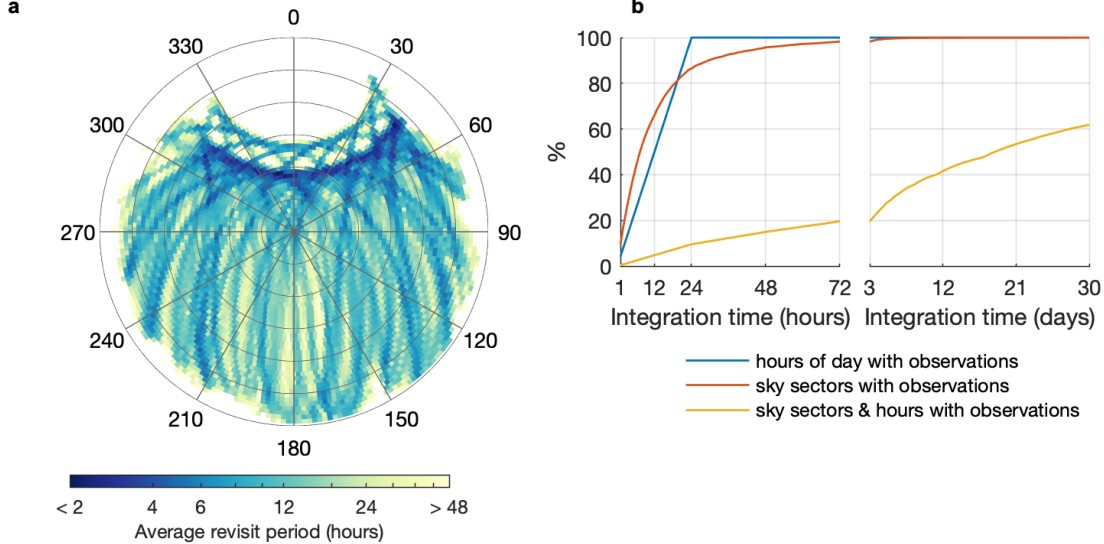


**Figure 6. (a) Sky plot illustrating the mean revisit time (average number of hours until the next overpass within a given sky sector). This includes the GPS, GLONASS, Galileo, and BeiDou constellations. (b) Average sampling statistics as a function of data integration time. For instance, 84% of all observable sky sectors are observed at least once after 24 hours of continuous measurements (red curve). After 30 days of continuous measurement, 62% of all observable sky sectors and hours of the day have**
**been observed at least once (yellow curve).**

Here, we propose to address this sampling problem by subtracting from each individual VOD observation

the long-term average (similar to what is shown in Fig. 5b) taken at the same azimuth and incidence angle.



The goal is to remove the static part of VOD, representing the uneven canopy distribution, and only retain anomalies from the average attenuation (Eq. 12). The long-term average at any given point of interest

(Eq. 13) is calculated inside a neighbourhood N that includes all measurements within some chosen angular distance $\delta$ from that point of interest (Eq. 14).

$$VOD^{anom}_{\varphi_i,\theta_i,t_i} = VOD_{\varphi_i,\theta_i,t_i} - VOD^{mean}_{\varphi_i,\theta_i} \qquad (12)$$

$$VOD^{mean}_{\varphi_i,\theta_i} = \frac{1}{n}\sum_{j=1}^{n}\left(VOD_{\varphi_j \in N_i,\, \theta_j \in N_i}\right) \qquad (13)$$

$$N_i: \quad \mathrm{hav}(\lambda_i - \lambda_j) + \cos(\lambda_i)\cos(\lambda_j)\,\mathrm{hav}(\varphi_i - \varphi_j) < \mathrm{hav}(\delta) \qquad (14)$$

where the terms $\varphi_i$, $\lambda_i$, and $t_i$ represent the azimuth, elevation, and time step of the point of interest (for angles expressed in degrees, $\lambda = 90 - \theta$). Equation 12 is the condition that determines if measurements belong to the neighbourhood around the point of interest '$i$'. The left term in Eq. 14 is the formula for the

haversine of the angle between any two points on a sphere. The right term is the haversine of a chosen angle $\delta$ which defines the extent of the neighbourhood.

An adequate value for $\delta$ may be selected based on the autocorrelation of the VOD observations with respect to the angular distance (Supplementary Fig. 1). This suggest that there is a high consistency of the

VOD estimates up to an angular distance of about 0.5 to 1°. Supplementary Fig. 1 also provides some indication of the repeatability of the measurements when taken at an interval of several days. As can be expected, observations separated by a longer temporal interval are in lower agreement. The selection of $\delta$ is ultimately a compromise between obtaining an accurate long-term VOD average while still retaining enough observations within the neighbourhood. In our case, we found that $\delta = 0.5°$ seems to be an

adequate value. To avoid excessive computations, we calculate $VOD^{mean}_{\varphi_i,\theta_i}$ at each node of a fine hemispherical grid with a spacing of 0.1°. The $VOD^{mean}_{\varphi_i,\theta_i}$ value closest to each individual $VOD_{\varphi_i,\theta_i,t_i}$ is then used in Eq. 12. Binning the calculated VOD anomalies into hourly averages, we produce a processed time series of the average VOD (Fig. 7, 'VOD processed'). To preserve the original absolute level of VOD, the average $VOD^{mean}_{\varphi_i,\theta_i}$ (across all $\varphi$ and $\theta$) is added back to the anomaly time series (otherwise the





'VOD processed' time series would be centred around zero). The serial autocorrelation of that processed

time series (Fig. 7d) is now dominated by a more credible 24-hour cycle.

**Figure 7. (a) Hourly time series of VOD before and after reducing the impact of irregular sampling caused by the GNSS orbit patterns. (b) Zoom on August, showing sub-daily variability in VOD. (c-d) Serial autocorrelation of the raw and processed VOD**
**time series.**



## 4. GNSS-based vegetation optical depth

### 4.1 Seasonal changes

In Fig. 8, we compare processed daily VOD averages against other observations. Using quality-checked satellite images from Sentinel 2 (Claverie et al., 2018), we calculate the enhanced vegetation index (EVI)
(Liu and Huete, 1995) at our site (Fig. 8a). EVI is a commonly used vegetation index and an overall proxy for vegetation greenness, health, and photosynthetic activity. Generally, we find that the temporal evolution of VOD appears to lag behind that of EVI by about 2 months. This is consistent with previous findings over drylands by Tian et al. (2016) who found a temporal shift (increasing as a function of forest density) between satellite-based VOD and vegetation greenness. They suggest that this may be explained
by the longer growing season and later peak time of woody plants compared to the herbaceous understory. Similar lags between peak NDVI and peak VOD have been observed in other regions of the world from satellite data (Wang et al., 2020;Tian et al., 2018). The peak EVI in June coincides with the maximum in available solar energy (Fig. 8b), and with an increase in VOD which could suggest a build-up of biomass in the canopy. This is followed by a slow decline in EVI which does not occur in VOD until the end of
August. The gradual decline in vegetation activity and health over the summer is typical of the region, and mainly a response to the overall increase in water stress resulting from warmer temperatures, drier atmosphere, and low soil moisture after months with no rainfall. The 2020 summer culminated with a record-breaking heatwave on September 6 (Fig. 8c), followed by a steady decline in VOD during fall season where some minor shedding of leaves could be observed at the site.





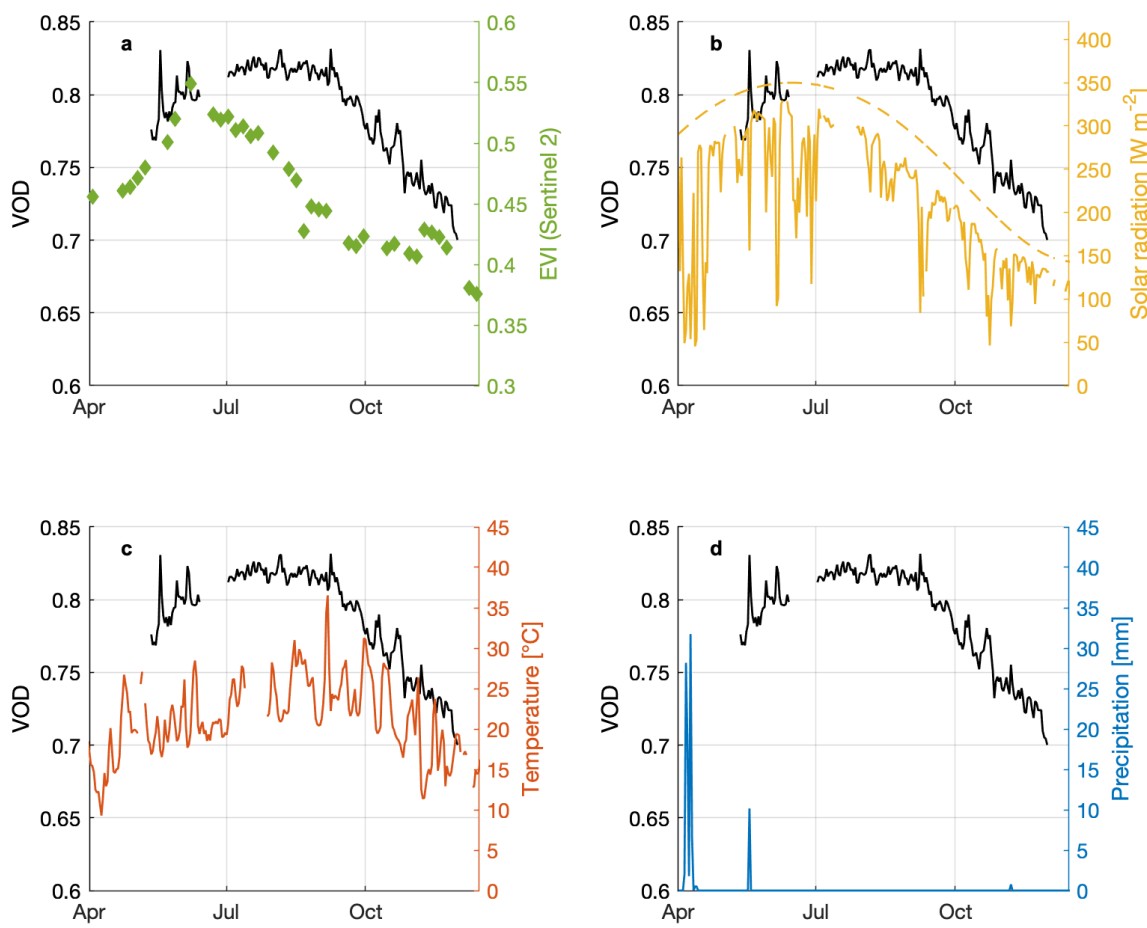


**Figure 8. Daily time series of GNSS-based VOD compared against (a) Enhanced Vegetation Index calculated from harmonized Sentinel 2 images (HLS v1.4, 30 meter resolution, https://hls.gsfc.nasa.gov/), (b) Solar and potential solar radiation observed at the reference site, (c) Air temperature observed at the reference site, and (d) precipitation totals measured at the closest rain gauge station.**

### 4.2 Diurnal cycle

Even though processed hourly VOD time series contain a certain amount of noise (e.g. Fig. 7b), they also show a relatively strong 24-hour cycle (Fig. 7d). One way of obtaining a more robust and precise estimate of that diurnal cycle is to calculate the average diurnal cycle from data aggregated over a long period of time, for instance several days or even a whole month. This is what is done in Fig. 9a. The average diurnal cycle of VOD is consistent with what would be expected from the perspective of plant physiology and its response to water stress over the course of a typical day. VOD culminates in the early hours of the morning, at around 5 to 6 AM, indicating a relatively "water-rich" canopy, because leaves and stems have







been replenished with water overnight, and likely also due to the occasional presence of dew in the canopy. This peak is followed by a gradual drop after sunrise (at about 6AM), concurrent with the onset of photosynthesis and transpiration, as well as an increase in vapour pressure deficit (i.e. an increase in atmospheric water demand). As a result, dew quickly evaporates and the vegetation starts losing more water through transpiration, thus depleting canopy water content. Around noon, some equilibrium is reached between plant water losses and plant water supply so that VOD becomes relatively stable. In the evening, plant rehydration causes VOD to rise back.

A minor peak in VOD is observed at around 3pm, but remains difficult to explain without further evidence. That peak might indicate a brief period of canopy rehydration, documented for instance in Douglas firs (Cermak et al., 2007, their Figure 6), resulting from midday stomatal closure (Xiao et al., 2021). The associated "midday depression" of transpiration and photosynthetic rates has been widely documented (e.g. Faria et al., 1996;Kamakura et al., 2011). However, theory (Fig. 2b) also predicts that VOD could slightly increase in response to higher canopy temperature, which would peak at about that time of the day. In Section 5.2, we present an attempt to disentangle these two possible contributions.

Overall, our results agree with previous observations of a diurnal cycle in VOD (e.g. Konings et al., 2017b;Holtzman et al., 2021;Vermunt et al., 2021;Prigent et al., 2022). Such diurnal VOD changes are consistent with our knowledge of canopy water storage dynamics, as derived from either continuous direct measurements (Zhou et al., 2018), or from the imbalance between plant water losses (i.e. transpiration) and plant water supply (i.e. measured with sap flow sensors) (Kocher et al., 2013;Cermak et al., 2007). The leaf samples collected on the site in October also confirm that some intra-day variability exists in relative leaf water content (Fig. 9b). Monitoring the diurnal cycle of plant water status is interesting because it can provide key information on plant hydraulic traits (Konings and Gentine, 2016) and enables disentangling the effects of limitations in root water uptake, plant transpiration, and water redistribution within the plant (Konings et al., 2021). In Fig. 9c, we investigate whether our method would be able to monitor such physiologically-relevant changes, and in particular here, seasonal changes in pre-dawn versus midday water status. We find midday VOD to be almost always lower than pre-dawn VOD, a behaviour that is entirely consistent with field observations of pre-dawn and midday leaf water potentials



(Martínez-Vilalta et al., 2014). Seasonally, both pre-dawn and midday VOD start to decrease in September with the start of a very dry period during which VPD remains high even during the night. Pre-dawn and midday VOD are significantly correlated with each other (r = 0.79), as is frequently observed with leaf water potential, however, we note that this might also occur (at least partly) because VOD is

sensitive not only to relative changes in water content but also to potential seasonal changes in the absolute amount of biomass present in the canopy (Momen et al., 2017). These mixed contributions from both absolute biomass and its relative water content are even better illustrated in Fig. 9d, where we find that the amplitude of the diurnal cycle in VOD becomes larger as denser sections of the canopy are considered.



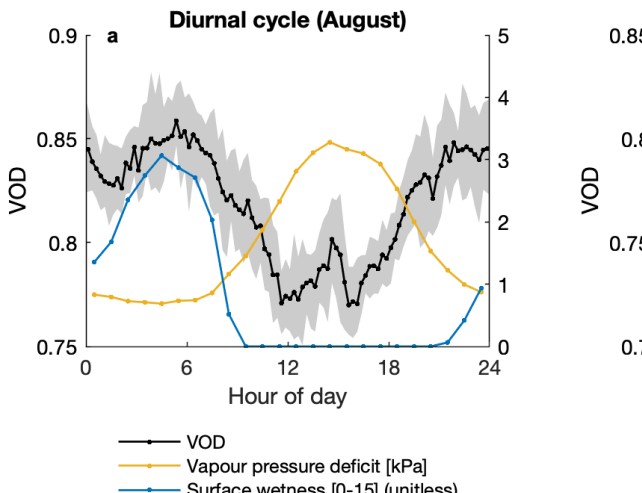
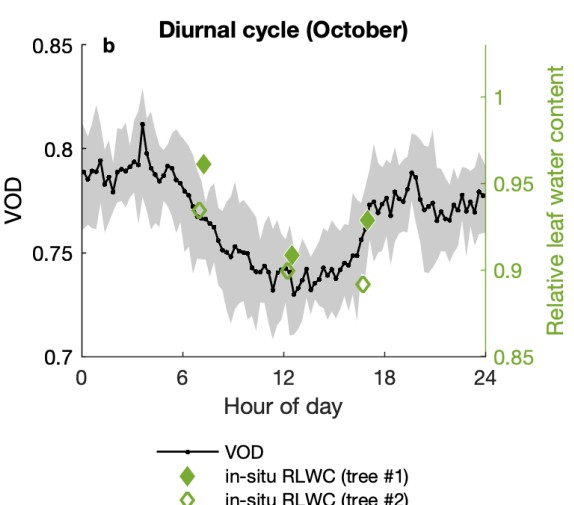

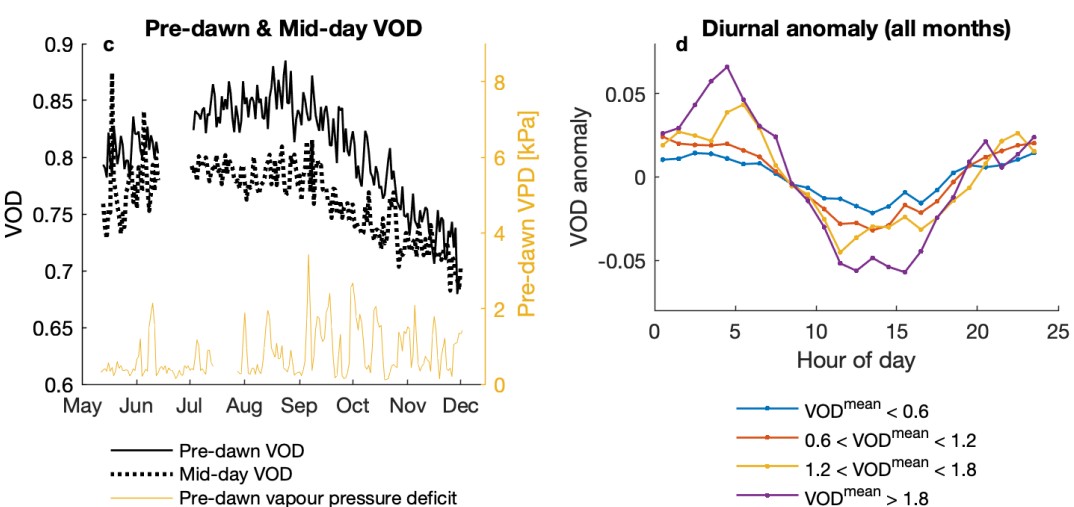


**Figure 9. (a) Average diurnal cycle for the month of August. Average VOD (black) is shown at a 15-minute sampling rate with shaded areas delineating the 25th and 75th percentiles. VPD and surface wetness data from the TCCON weather station have hourly resolution. (b) Average diurnal cycle for the month of October compared with in-situ measurements of relative leaf water content. (c) Daily pre-dawn and midday VOD, calculated using all observations within the window 4AM-6AM and 12PM-2PM respectively.**

**(d) Diurnal VOD anomaly (centred around zero) measured for progressively denser classes of canopy (based on the long-term VOD average, see Fig. 5b).**



## 5.  Retrieval of canopy density and water content

### 5.1 Approach and algorithm

In the following, we demonstrate an approach to retrieve changes in canopy density and water content at

hourly resolution based on the transmissivity model presented in section 2.3. Combining Eq. 1, 4, 5, and

8 we obtain the following expression for modelled $\widehat{VOD}$:

$$\widehat{VOD}_t = 2k_e h = 2\frac{2\pi}{\lambda_0}\left[-\Im\{\sqrt{\varepsilon_{veg_t}}\}\right]v_{veg_t}\,h \qquad (15)$$

Canopy height $h$ is 7 meters, and $\lambda_0$ is the free-space wavelength. This leaves as free parameters: $v_{veg_t}$,

the time-dependent volume density of the canopy (in m³/m³), and $\varepsilon_{veg_t}$, the time-dependent bulk

dielectric constant of the vegetation, which is itself a function of the (measured) temperature, (unknown)

water content ($m_g$) and (unknown) salinity of the plant water (Ulaby and El-rayes, 1987). A similar

expression may be found for instance in Kerr and Wigneron (1995) or Guglielmetti et al. (2007).

Here, we use the canopy-averaged processed VOD time series (i.e., Fig. 7b) as the observed VOD. This

means that canopy density and water content are assumed to evolve homogeneously over the whole

canopy. If the forest canopy is very heterogeneous, this might not be a suitable approximation. For

instance, some groups of trees may evolve at different speeds, or individual trees may exhibit different

responses to water stress. If this is suspected, a retrieval may be performed for each individual tree,

potentially at the cost of retrieval accuracy since less observations will be available. In practice, this would

mean separating the field of view of the antenna into sub sectors (for instance based on data similar to

Fig. 4) and computing a processed VOD time series for each sub sector. However, and for the sake of

simplicity, this is not done here, and we perform only one retrieval for the whole canopy.

Disentangling the effects of changes in overall biomass density ($v_{veg}$) versus variations of its water

content ($m_g$) is one of the main challenges when trying to interpret and better understand VOD (Momen

et al., 2017;Konings et al., 2019). Because changes in biomass tend to unfold at a much slower pace than

changes in relative water content, a common strategy has been to assume that long-term changes in VOD

are mostly related to biomass, while short-term changes (and especially a diurnal signal) are most likely

due to variations in water content (e.g. Konings et al., 2016). In the retrieval detailed below, we will make

use of this assumption and allow $v_{veg}$ to contain only low-frequency (long-term) changes. We also




assume that the temperature of the canopy can be approximated by the air temperature. While leaf and air

temperatures are not necessarily equal, as documented by many field studies, an error of a few degrees is

negligible for the purpose of calculating $\varepsilon_{veg}$ (Fig. 2b) and our main purpose is only to gain a reasonable

representation of the seasonal and diurnal effects of temperature on the dielectric constant. We also note

that these temperature impacts on the dielectric are often neglected in other studies.

Finally, the plant water salinity is also unknown in our case, but most likely within a range of about 1 to

11 psu according to previous experience (Ulaby and Long, 2014). In the absence of any other data, we

assume that salinity is constant over the whole time period. In our retrieval algorithm, a range of a priori

values for vegetation water salinity is tested and the value yielding the best overall (season) fit to the

observed VOD data is selected as the most plausible. We note that salinity, in the context of the dielectric

model of Ulaby and El-rayes (1987), is meant to account for the ionic conductivity of the plant water (due

to both sugars and salts). Thus, the salinity yielding the best fit might not necessarily reflect the actual

(NaCl) salinity of the plant water.

Below, we summarize the retrieval algorithm step by step. The root mean square error between modelled

and observed VOD is used as the cost function and optimization is carried with a simplex search method.

We define the search space for $m_g$, the gravimetric moisture content (water mass / fresh mass), as [0.3,

0.7]. This is guided by the average values measured at the site ($m_g$= 0.45 ± 0.02 g g$^{-1}$), and also by data

from Scoffoni et al. (2014) for *quercus agrifolia* in the Los Angeles area ($m_g$= 0.48 g g$^{-1}$). The search

space for $v_{veg}$ (volume of vegetation material per m$^3$ of canopy) is loosely defined as [0.0001, 0.01 m$^3$

m$^{-3}$] based on the indications of Ulaby and Long (2014).

Algorithm

1. Select an a priori value for salinity.

2. For each 24-hour period, optimize $v_{veg}$ (1 value for the entire 24-hour period) together with $m_g$

   (24 hourly values). The result is a time series of daily $k$ and hourly $m_g$ covering the whole season

   (Fig. 10a).



3. Filter the time series of $v_{veg}$ with a low-pass filter. Here we use a local regression filter (LOESS) with ±30 days width (Fig. 10a).

4. Optimize $m_g$ using the new $v_{veg}$ values obtained at step #3 (Fig. 10b).

5. Filter the time series of $m_g$ to reduce the high-frequency noise. Here we use a LOESS filter with ±2 hours width (Fig. 10b).

6. Evaluate the agreement between the modelled and observed canopy-averaged VOD time series (with $v_{veg}$ and $m_g$ from steps #3 and #5) and determine an optimal salinity (Fig. 10c).

Because $v_{veg}$ is kept constant only for the duration of a day, there is some high-frequency variability in the $v_{veg}$ time series that is obtained after step #2 (Fig. 10a). These sudden and unrealistic jumps of $v_{veg}$

also contaminate the estimates of $m_g$ (not shown). These problems are alleviated in step #3, where daily estimates of $v_{veg}$ are low-pass filtered, consistent with the assumption that changes in biomass usually occur at a relatively slow pace. A new hourly time series of $m_g$ is then obtained based on the filtered $v_{veg}$ time series in step #4 (Fig. 10b). Because the $m_g$ time series is contaminated by some noise inherited from the VOD observations, some mild smoothing is applied to $m_g$ in step #5 (Fig. 10b). Steps #2 to #5

are repeated for different values of salinity and we retain the optimum of the cost function as the most likely value (Fig. 10c). Here we find an optimum with a salinity of 8.9 psu, which is a physically plausible value.





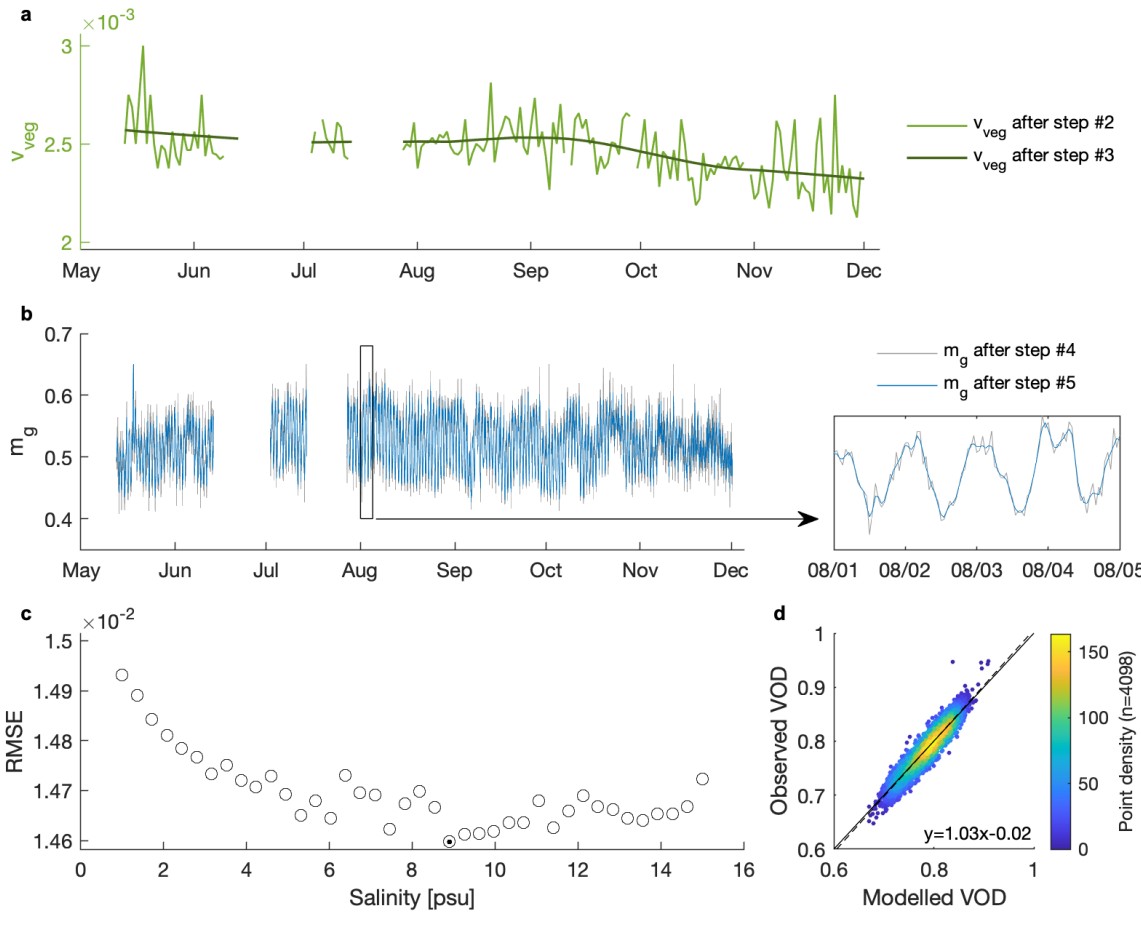

**Figure 10. (a) Time series of $v_{veg}$ after steps #2 and #3. (b) Time series of $m_g$ after steps #4 and #5 with a zoom on a short period**
**for better visibility. (c) Root mean squared error obtained for various salinity values. (d) Scatter plot of the modelled versus observed VOD. The data shown in (a), (b) and (d) is based on a salinity of 8.9 psu.**

Before we interpret these results, some limitations to the presented approach need to be emphasized. First,

the dielectric model of Ulaby and El-rayes (1987) was originally developed for leaves, however, it is clear

that branches and stems may also contribute to canopy extinction. To our advantage however, Kurum et

al. (2009) have determined with numerical simulations (at L-band) that leaves have a significant impact

on extinction, while branches have a dominant contribution only in terms of backscatter, and trunks have

a negligible impact on extinction. Steele-Dunne et al. (2012) arrived at similar conclusions and concluded

that leaf moisture is by far the dominant control on vegetation transmissivity at L-band (for both





polarizations). Observations by Mätzler (1994) in a deciduous forest also showed a clear dependence of the transmissivity to the presence or absence of leaves in the canopy. Thus, we assume that the dielectric model of leaves and its sensitivity to moisture, temperature, and salinity provides a good approximation for the behaviour of the whole crown (branches and stems included). However, the impact of intercepted water (due to dew deposition or rainfall) is not explicitly represented and will thus be compensated for by

an overestimation of the gravimetric moisture content. There is unfortunately not enough empirical data in our case to also retrieve intercepted water independently.

### 5.2 Results and interpretation

In Fig. 11a, the retrieval of gravimetric moisture content ($m_g$) is compared against observations of leaves taken at the site on October 18. There is a bias of 0.04 g g$^{-1}$ between the retrieval and the observations, a

surprisingly good performance given the assumptions made during the retrieval of $m_g$, and the fact that a few leaf samples are not necessarily representative of the entire canopy. The relative difference between dawn and daytime values (about 0.03 g g$^{-1}$) is consistent between the retrieval and the observations. We also find that the relationship between retrieved hourly $m_g$ and VPD becomes narrower (Fig. 11b) compared to the relationship between hourly VOD and VPD (Fig. 11c). Even though this does not provide

any formal validation of the $m_g$ retrieval, it does suggest that the retrieval is somewhat successful in concentrating in $m_g$ a response to atmospheric water demand that is consistent with observed plant stomatal behaviour (e.g. Grossiord et al., 2020).

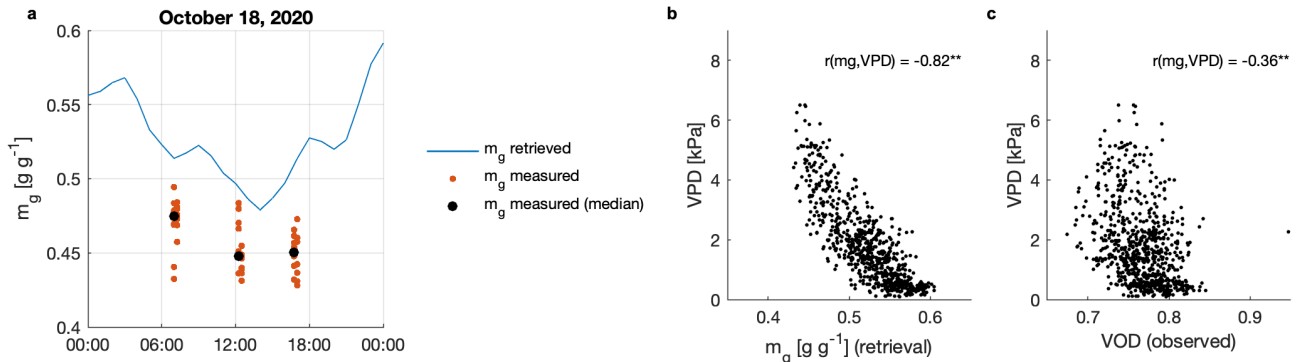





**Figure 11. Retrieved and measured values of gravimetric moisture content ($m_g$) on October 18 (a). Scatter plots of the hourly values of $m_g$ (b) and VOD (c) against vapour pressure deficit (VPD), for the month of October (both correlations significant at p<0.01).**

Dry aboveground biomass (AGB, kg m⁻²) can be calculated by multiplying the retrieved volume of the vegetation material ($v_{veg} \cdot h$, m³m⁻²) with the average density of the dry leaf material ($\rho_{dry}$). Here we use

leaf density to remain consistent with the model's assumptions, but it is unknown if the density of leaves, branches, wood, or some weighted average of all three would be more relevant at this stage. We use a value of $\rho_{dry} = 630$ kg m⁻³ reported by Scoffoni et al. (2014) for *quercus agrifolia*.

$$AGB = v_{veg}\, h\, \rho_{dry} \qquad (16)$$


Canopy water content (CWC) is then calculated as follows.

$$m_g = \frac{water\ weight}{fresh\ weight} = \frac{CWC}{CWC+AGB} \qquad (17)$$

$$CWC = \frac{AGB\ m_g}{(1-m_g)} \qquad (18)$$


The resulting time series are shown in Fig. 12a. For AGB, we obtain a mean value of 10.9 kg m⁻² with very little seasonal variations, as may be expected for an evergreen forest. Because in situ estimates of AGB are not available at our experimental site, we can only put this value in context with the literature. Several studies have empirically linked VOD observations to global AGB datasets that are based on

satellite data and forest inventories (Avitabile et al., 2016). For instance, using the exponential relationship calibrated at L-band by Vittucci et al. (2019), we obtain (for an average VOD value of 0.79 at our site) an AGB of 13.8 kg m⁻², not so far from our estimate. Other results from Brandt et al. (2018) suggest a linear relationship between AGB and L-band VOD, with a sensitivity of about 133 Mg ha⁻¹ per unit VOD, which would yield an AGB of 10.5 kg m⁻² at our site. Thus, existing empirical relationships

between VOD and AGB would suggest that the results obtained with the retrieval algorithm and its simple but physics-driven model are reasonable.





In terms of canopy water content (CWC), which is a function of both $m_g$ and AGB, we find a long-term mean of 12.1 kg m$^{-2}$ at our site. Comparisons with other studies are quite difficult here because the relationship between VOD and vegetation water content is poorly known for forests. Many studies over

grasslands and croplands have demonstrated an empirical linear relationship of the form $VOD = b \cdot CWC$. However, b is also known to be vegetation- and time-dependent (Jackson and Schmugge, 1991;Van de Griend and Wigneron, 2004) and it has been argued that such a linear relationship is not necessarily appropriate for forests (Kurum et al., 2012;Le Vine and Karam, 1996). A mean CWC of 12.0 kg m$^{-2}$ is in broad agreement with the few studies which measured forest CWC. Recently, Kurum et al. (2021)

reported 7.3 – 25.6 kg m$^{-2}$ across various plots of a deciduous broadleaf forest in Manitoba (Canada) with mean height of 10.9 m. Yilmaz et al. (2008) estimated CWC values ranging from about 2 to 10 kg m$^{-2}$ for a deciduous forest in Iowa (USA). In the SMAP soil moisture retrieval algorithm, vegetation water content values are estimated from NDVI (Chan et al., 2013). Their resulting (unvalidated) global map suggests a range of 6 to 18 kg m$^{-2}$ for various types of forest biomes.

In Fig. 12b-c, we investigate the temporal consistency (over the whole season) between the retrieved CWC values at our site and satellite observations of the Normalized Difference Water Index (NDWI) from Sentinel 2 (Gao, 1996;Claverie et al., 2018). NDWI is a good proxy for vegetation water content and is based on optical and near-infrared measurements, thus providing fully independent observations with respect to our retrieval. There are 24 days when NDWI observations are available, not flagged for

cloud cover, and concurrent with a CWC estimate (Fig. 12b). We find a relatively good agreement between CWC and NDWI (r = 0.70), higher than the agreement between VOD and NDWI (r = 0.53). Interestingly, this agreement is quite dependent on the timing of the in-situ CWC (or VOD) measurement (Fig. 12c). For instance, comparing the 10:30 AM NDWI with the 12:30 PM CWC (or VOD) would yield a substantially lower (and statistically non-significant) correlation of r = 0.29 (0.30 for VOD). This time-

dependence of the agreement is a good indication that the diurnal cycle of CWC is well captured and that VOD alone is not fully representing its dynamics.

The diurnal dynamics of CWC are particularly strong, with a diurnal amplitude of 3.8 kg m$^{-2}$ on average, meaning that 28% of the pre-dawn CWC is depleted over the course of the day (Fig. 12d). Such diurnal variations may seem important for a forested ecosystem but are not entirely inconsistent with other





studies. Over a corn field in the Netherlands, Vermunt et al. (2021) observed that midday vegetation water content was decreased by 10-20% on average compared to pre-dawn levels and even by 35.4% on a particularly warm day. The results of Mirfenderesgi et al. (2016), who investigated the transpiration and sap flow dynamics of oaks in New Jersey with a hydrodynamic model, suggest a diurnal amplitude of about 15% for just stem water storage. Matheny et al. (2017) also reconstructed a diurnal amplitude of

14.6% to 22.3% of the stem water storage from in situ sap flow measurements of red maple in Michigan. Since our estimate of CWC also likely incorporates an additional contribution from dew (discussed in the next section), an average CWC amplitude of 28% would not be unexpected.

We find that our retrieval of CWC is slightly lagged compared to the diurnal average VOD cycle (Fig.

12d). This lag is due to the diurnal cycle of temperature and its effect on the dielectric loss (and thus on VOD), as represented in the dielectric model (Fig. 2b). The effect can also be seen in Fig. 10e which focuses on the month of August. Here a minor peak in VOD can be observed at about 3PM, in the centre of the midday depression (also see Fig. 8a). Our retrieval suggests that at least some of this peak is in fact not related to rehydration but to a peak in diurnal temperature. In Fig. 13, we use the transmissivity model

(Eq. 15) to explore the influence of moisture and temperature on modelled VOD (by enforcing a seasonally constant value for temperature and gravimetric moisture respectively). As temperature increases during the day, it increases the dielectric loss and leads to a higher modelled VOD (Fig 13a), thus counteracting the effect of moisture content changes to some extent. This effect was particularly pronounced during the heatwave which struck the area from September 5th to 6th (Fig 13b). Here it can be

seen that there is relatively little response in terms of VOD during the heatwave, with even a minor increase on September 5. Because it is taking the response of the dielectric loss to temperature into account, the retrieval algorithm needs to compensate the effect of temperature with a marked CWC drop over that period (e.g. see Fig. 10a), which makes sense from a physiological standpoint. Indeed, the record-breaking heatwave was accompanied by VPD values of up to 8kPa on both days (compared to an

average of 1kPa the week before) and it would be very unlikely to see no response of CWC (and VOD) to such a level of stress.





This modelled response of VOD to high temperatures emerges directly from the dielectric model of Ulaby and El-rayes (1987). It is also predicted by another semi-empirical dielectric model of leaves proposed by Matzler (1994), who confirmed such a temperature dependency with seasonal observations made in a
beech forest (although they did not control for potential moisture changes as a covariate). It is very important to note that, at L-band, the magnitude as well as the direction of the dielectric loss's dependence on temperature is dependent on the temperature itself (Fig. 2b). For instance, at temperatures between 0 and 20°C the sensitivity of L-band VOD to temperature is negative (Schwank et al., 2021). We note that this temperature dependency can be quite crucial when interpreting water stress from VOD
measurements. As water stress conditions often correlate with warm temperatures, one must be careful in interpreting VOD time-series over a large dynamic range of canopy temperatures as VOD alone might lead to unphysical interpretations, in our case an increase of canopy water in the middle of a heat wave.



**Figure 12. (a) Retrieval of canopy water content (CWC) and dry aboveground biomass (AGB). (b) Comparison between NDWI estimated from Sentinel 2 (HLS v1.4, 30 meter resolution, *https://hls.gsfc.nasa.gov/*) and our retrieval of CWC at the same hour as the Sentinel 2 overpass. (c) Correlation between Sentinel 2 NDWI and CWC/VOD observed at different hours of the day. (d) Long-term average diurnal cycle of CWC and VOD (as predicted in Eq. 15). (e) Same as (d) but for the month of August only.**



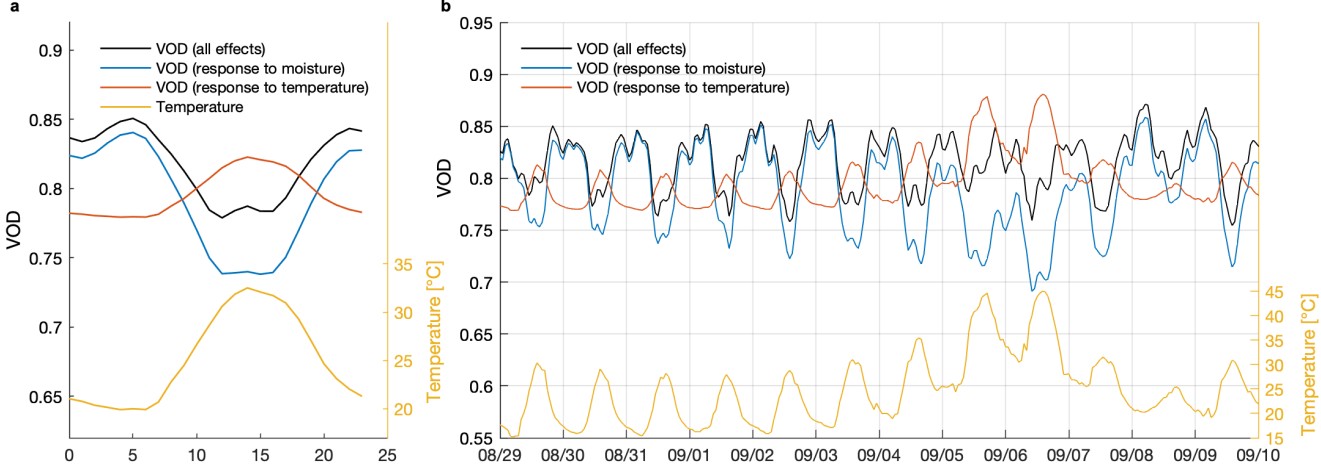

**Figure 13. Counteracting effects of leaf gravimetric moisture and temperature on diurnal VOD changes, as predicted by the vegetation dielectric model of Ulaby and El-rayes (1987). (a) Modelled diurnal VOD response to leaf moisture and temperature, averaged over the month of August. The VOD response to gravimetric moisture is estimated by setting temperature to its mean value in the model (and inversely for the response to temperature) (b) Modelled hourly VOD response to moisture and temperature in the context of the record-breaking heatwave of September 5th to 6th.**

## 5.3 Rainfall interception and dew

The only significant rainfall event during the measurement period occurred on May 18th (Fig. 14a). A cumulated precipitation amount of 10.2 mm was measured on that day which coincided with an increase in VOD of about 20% compared to the usual diurnal cycle. Within a few hours following rainfall, this excess VOD quickly subsided but did not fully disappear until the day after. These results indicate that L-band VOD is quite sensitive to intercepted rainfall, as could already be inferred from other studies on X-band VOD involving either satellite observations (Xu et al., 2021) or in-situ radiometer data (Schneebeli et al., 2011). The VOD anomaly also lasted longer compared to the surface wetness measurements taken at the reference station (Fig. 14a). This is likely because the surface wetness instrument is exposed to sunlight and an open atmosphere, such that its surface water evaporates much faster compared to what happens in a forest canopy. If future research at eddy-covariance tower sites can demonstrate that such VOD measurements are a good proxy for intercepted water, this may provide a useful constraint to the partitioning of evapotranspiration fluxes into different sources (i.e. evaporation versus transpiration).





For completeness, we also investigate the effect of dew on our measurements in Fig. 14b-c. Unlike rainfall events, dew events have smaller impacts on VOD and are more difficult to isolate from other sources of variability. Thus, we focus on the retrieved CWC time series, as it is less influenced by day-to-day variability in temperature compared to the VOD measurements (as discussed in the previous section). To diagnose some possible effects of dew on CWC, we separate a two-month observational subset into two samples based on the daily maximum relative humidity ($RH_{max}$). We use a threshold of 70% relative humidity to distinguish nights with and without a potential for dew formation (Ritter et al., 2019). We then calculate the average diurnal CWC cycles for each of these two subsets (Fig. 14b) and compare their relative difference against the surface wetness measurements taken at the reference site (Fig. 14c). Because the surface wetness sensor tends to saturate relatively quickly, we exclude nights where the wetness sensor is stuck at its maximum value. While this does remove some nights where a lot of dew deposition is occurring, it has the advantage of preserving the proportionality between CWC and the wetness sensor measurements. We find a relatively good agreement between CWC and the wetness sensor data (Fig. 14c), suggesting that dew deposition is reflected in our CWC retrieval. This finding is of importance since dew deposition, even in southern California, would then influence and potentially bias calculated differences between midnight and mid-day satellite-based VOD, which have been used to interpret plant hydraulic behaviour (Konings and Gentine, 2016). The two-peaked structure in Fig. 14c seems to be arising from the combination of individual dew accumulation events occurring either after sunset or before sunrise. It is important to note that because the transmissivity model does not represent surface water in a dedicated way (Schneebeli et al., 2011), the data in Fig. 14c should not be used to derive actual dew amounts. Here we can only conclude that L-band VOD is likely influenced by dew deposition, unlike Holtzman et al. (2021) who found no evidence for the impact of dew on VOD measurements in an oak forest, but in agreement with conclusions from several other studies (Xu et al., 2021;Schneebeli et al., 2011;Khabbazan et al., 2022).





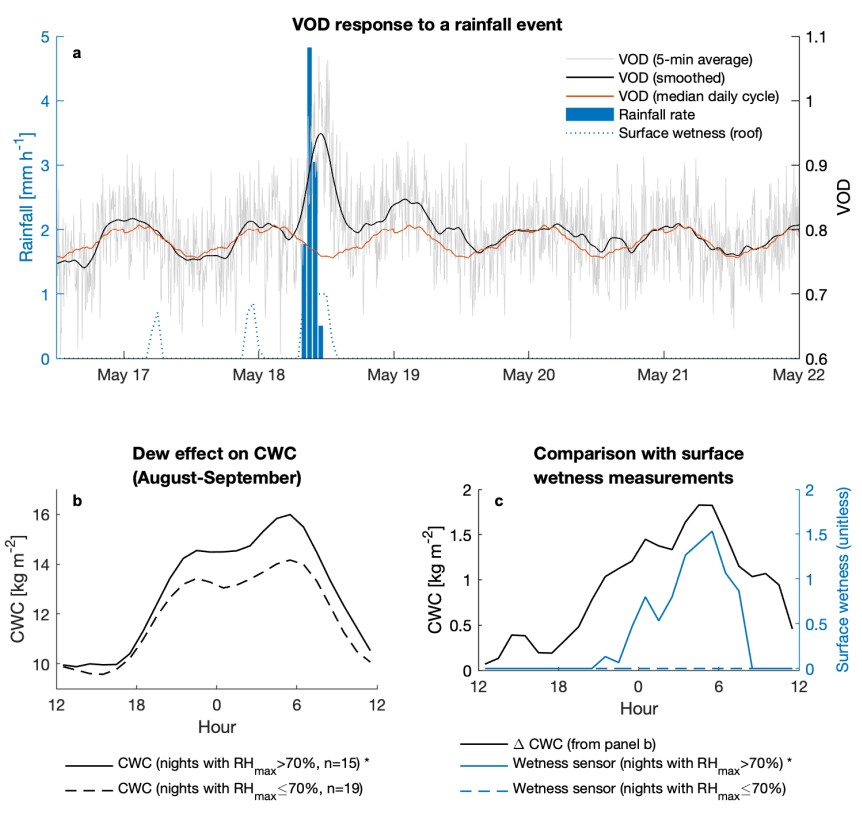

**Figure 14. Surface canopy water effects. (a) VOD response to a rainfall event. The 5-minute time series are smoothed with a LOESS filter (±1.5 hour width). (b) Average diurnal CWC for nights with and without conditions favourable to dew deposition (solid versus dashed line). (c) Difference in CWC based on (b) compared against wetness sensor measurements (taken at the reference site).**


## 6. Conclusions

In this paper we have demonstrated that a pair of GNSS receivers can be used to continuously measure L-band VOD in a forest stand. Thanks to the diversity of GNSS satellite orbits, a hemispheric scan of the canopy can be obtained, offering the opportunity to individually monitor specific trees, groups of trees, 700 or classes of canopy density. While continuous changes in GNSS orbit patterns and constellation configurations complicate the analysis of raw observations, we provide here a relatively straightforward solution to alleviate this problem and produce credible VOD time series. Pooling observations from the four largest GNSS constellations (GPS, GLONASS, BeiDou, and Galileo), we show that VOD anomalies can be resolved at hourly resolution. In particular, our approach is able to identify a diurnal cycle in VOD





which appears consistent with what has been reported in previous recent studies (Holtzman et al., 2021;Vermunt et al., 2021;Konings et al., 2017b).

In a further step, we use existing models of canopy microwave transmissivity and vegetation dielectric parameters (Ulaby and Long, 2014) to demonstrate the feasibility of using such GNSS-based VOD observations to derive information on canopy water content and aboveground dry biomass. Owing to the

limited number of ancillary in situ measurements, this retrieval algorithm was not evaluated against long-term ground truth observations and should be seen as a proof-of-concept. Still, the resulting dry aboveground biomass and CWC estimates agree with what has been reported in similar studies. In addition, we show that the CWC estimates are in good agreement with satellite observations of the research site (NDWI from Sentinel 2), but only if the hourly CWC estimates are taken precisely at the

time of the satellite overpass. This dependency of the agreement on the observation timing provides convincing and independent evidence that the CWC time series derived from GNSS-based VOD contains valuable information. We also investigate the potential effects of diurnal changes in temperature on the dielectric constant of the vegetation and its impact on L-band VOD and the retrieved CWC. We show that temperature effects on the vegetation dielectric at L-band are predicted to play a minor but non-

negligible role and can modulate VOD variability, especially for the case of diurnal variability as well as during extreme events such as heatwaves. Our results suggest that diurnal variability in VOD due to water content could be dampened and in some cases even be reverted by the variability in temperature. This is because temperatures > 20°C cause an increase of the dielectric loss in saline water, causing VOD to increase over the day, in a direction opposite to the expected effect of diurnal canopy dehydration. The

magnitude of this temperature effect is dependent on the microwave frequency and the ionic conductivity of the plant water according to the empirical model of Ulaby and El-rayes (1987). Finally, we provide some evidence that GNSS-based VOD is sensitive to surface canopy water, for the cases of both rainfall interception and dew deposition.

Future work may focus on various aspects. Future deployments of GNSS-based measuring systems like the one proposed here will be made at heavily monitored ecohydrological research sites in order to facilitate cross-comparisons with other in situ data. At the time of writing, we have recently equipped





three forested FluxNet eddy-covariance sites with pairs of GNSS sensors, one in the U.S. (US-MOz) and two in Switzerland (CH-Lae and CH-Dav). Based on these new deployments, we provide below a few

recommendations. The total cost to equip an existing research site was of about 2000 USD, with two thirds of that amount dedicated to acquiring the scientific instruments. The reference instrument may be placed at the top of a flux tower or at some other close location (i.e. < 5 km away) with the clearest possible view of the sky. Positioning the subcanopy antenna is not subject to strong restrictions. We suggest to place the antenna in direct view of frequently monitored trees and not too close to strong

reflectors (large tree trunks, buildings). The sampling characteristics discussed in section 3.2 and reported in Fig. 6a may be helpful in guiding new installations. Over flat terrain, the maximum extent of the observation footprint is dependent on the height of the vegetation (minus the subcanopy antenna height). Assuming that measurements with an elevation angle lower than 10° are discarded, the footprint may be roughly estimated as a circle with a radius of $r = h/\tan(10°)$, i.e. r ≈ 57 m for $h = 10$ m. We provide

further recommendations in Supplementary Table S1. Other potential future objectives may include an evaluation of GNSS-based VOD estimates against other VOD measurements made by a tower-mounted radar or radiometer. Even though our observations do not suggest it, we cannot yet exclude the possibility that GNSS-based VOD has some type of bias compared to these other types of instruments (which do not constitute a reference per se). The question of how to best process the GNSS data in order to obtain VOD

time series also remains to be explored. The initial solution provided in Section 3.3 may likely be refined to further reduce some of the noise still present in sub-hourly GNSS-based VOD time series. Some additional information may also be gained by using GNSS signals at multiple frequencies (we used here the most common signal, which is emitted at around 1.57 GHz, but individual constellations also broadcast signals at lower frequencies, up to 1.17 GHz).


The results presented here suggest that GNSS-based VOD may have the potential to fill a key research gap in terms of linking satellite-based L-band VOD observations to ground observations. This contribution could take place in several ways. For example, arrays of GNSS receivers deployed within the spatial footprint of a satellite VOD grid cell (i.e. about 30 km) may serve to estimate a regional average

VOD that would be suitable as ground truth for the satellite products. In addition, long-term in-situ VOD



observations performed at existing ecohydrological research sites may serve to develop and evaluate retrieval algorithms that aim to transform VOD into other relevant quantities of interest like aboveground biomass, canopy water content, or leaf water potential. To the benefit of these research sites, GNSS-based VOD provides a useful proxy to upscale and gap-fill time series of the time-consuming and labour-intensive measurements of leaf water status and biomass. The ability to detect rainfall interception and dew deposition at the scale of a whole canopy may also provide some key information to improve our understanding of water, energy, and carbon fluxes at these sites. While GNSS-based monitoring of the Earth system remains a relatively diverse and emerging research field, remote sensing of GNSS-based VOD appears as a particularly promising application, because of its ability to address a series of research objectives at a modest cost.

**Code and data availability**

Raw and processed GNSS data files are freely available upon request. Weather data is publicly available at http://tccon-weather.caltech.edu/ and https://dpw.lacounty.gov/wrd/rainfall/. Harmonized Sentinel-2 data is publicly available at https://hls.gsfc.nasa.gov/. The tecq software is publicly available at https://www.unavco.org/software/data-processing/teqc/teqc.html.

**Competing interests**

The authors declare that they have no conflict of interest.

**Author contributions**

V.H. developed the concept, conducted the measurements, and the analysis. Both authors contributed to the methodology, visualizations, and writing.

**Acknowledgments**

V.H. acknowledges support from the Swiss National Science Foundation (grants no. P400P2_180784 and P4P4P2_194464). We thank Brian Dorsey from the Huntington Library, Art Museum, and Botanical Gardens in San Marino, Los Angeles County for providing us with access to the gardens, participating in the measurements, and coordinating technical support for the research site. We thank Ken Hudnut, Evelyn



Roeloffs, and Aris G. Aspiotes from the United States Geological Survey for lending us the equipment needed to conduct preliminary investigations. We thank Kristine Larson from the University of Colorado for providing us with initial recommendations on the overall set-up.

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
