# Peer review of "Continuous ground monitoring of vegetation optical depth and water content with GPS signals"

_Biogeosciences, 2022_

## Referee Comment (RC1)

This paper uses a pair of two ground-based GNSS receivers to estimate L-band VOD in a forested field. Similar configuration has been previously tested by others. Unique part of this study is for collection of a field data over a period of 8 months with analysis to get insights into vegetation response in different time scales such as hours-minutes for intercepted rainfall and dew, days-hours for canopy water content, and months-weeks for canopy structure. I think that the topic is very important for the community and the approach is very promising to monitor trees characteristics nondestructively in a continuous fashion. However, I have several major concerns on the VOD estimates, the model, and assumptions as they are the basis for the analysis.

   (1) **Polarization**: The current setup measures the signal in RHCP-RHCP (RR) configuration, which might be significantly different from H-pol or V-pol VOD estimations from the satellites. This relation among various polarizations (RR, H or V pol) has not been established for VOD in the literature, so it is hard to interpret how RR-pol VOD estimates can be used for satellite data validation.
   (2) **Model**: The model used here is a dielectric mixing model, which is valid for electrically small constituents. At L-band, primary (even secondary) branches can violate the assumption. In addition, the model ignores the polarization, which could be significant for tree canopies with certain preferred oriented branches.
   (3) **Assumption**: The current model ignores volume scattering. As the data suggests (from my own experience as well), the signals at high elevation under canopy could be larger than the signal in open sky environment. This is a clear indication of volume scattering. The impact of volume scattering may vary from trees to trees and elevation angles to angles. Without physical modeling simulations for various scenarios, it would be difficult to develop methodologies to disentangle the direct signal from volume scattering.

Additional references (use a similar approach) to consider:
https://doi.org/10.1109/IGARSS47720.2021.9555155
https://doi.org/10.1155/2017/6941739

**Specific Comments:**

Page 1, Line 16: I think you meant "signal strength".

Page 1, Line 24: Have you checked the impact of rainfall and dew on the instrument? Any water on the antenna surface can change its radiation characteristics.

Page 2, Line 29: I think you meant "receiver systems".

Page 2, Line 40: I would say "continuously gathered network of ground truth data".

Page 2, Line 53: Replace "either or" with "both and" as both mentioned factors impact the transmissivity simultaneously.

Page 3, Equation (1) : Add polarization symbol to this equation since VOD and gamma depends on polarization.

Page 4, Footnote: Coarse spatial resolution is not due to low signal energy. It is due to incoherent nature of the signal that requires real aperture antenna (limited in size).

Page 5, Line 114: Please state the polarization of antennas used with these receivers here. In addition, receivers could be placed on the ground (without tripod) as long as they are leveled and cleared from the surrounding obstructions.

Page 7, Lines 149 - 151: The pictures of receiver set-up would be useful to add to the manuscript.

Page 7, Line 156: Please state the polarization of the antenna.

Page 8, Line 179: This model is "dielectric mixing model". The physical model implies either "discrete scatter" based radiative transfer or wave theory, none of which are used in this paper. I would suggest not to use term "physical" for the model.

Page 8, Line 181: This is a major assumption which is not true for well-developed trees. Tree branches scatter and attenuate significantly while leaves attenuate mostly at L-band, so the received signal is expected to be a blend of both volume scattering and attenuation, depending on the vegetation, polarization, and incidence angle.

Page 8, Equation (4): This is repetition of Equation (1). I would define VOD here, instead.

Page 8, Equation (5): This formula is applicable if inclusions are much smaller than wavelength as you correctly stated in the following paragraph. The formula can be applied to randomly oriented agricultural crops with no preferred scattering orientation and smaller than wavelength, but the validity region would be way off for tree canopies. The more descriptive formulation can be written as a function of forward scattering amplitudes of individual tree constituents. An example can be found at equation (8) of reference Guerriero et al. 2020.

Page 8, Line 196: That is correct that Guerriero et al. 2020 showed that coherent line-of-sight signal mostly dominates over volume scattering for RHCP-transmit RHPC-receive cases. However, it is hard to generalize this statement as this behavior can depend on the elevation, biomass, and type of trees. The question is if it is applicable to your case.

Page 9, Lines 231-214: This semi-empirical formula is developed for agricultural crops. It is not clear how it can be applied to trees, which are electrically much larger at L-band.

Page 9, Line 220: I would call the dielectric mixing model approach as "zero-order" as the first order has completely different meaning in physical models such as RTE or wave theory.

Page 11, Lines 248-249: I would say "yield information on VOD" as stated at page 5, Line 116 as I stated above difference SNR can include some of the volume scattering.

Page 12, Lines 260-261, Page 13, Lines 275-276: It is true that ground reflections would be small since your antenna is designed to reject reflected LHCP signals from the ground, but it would still receive volume scattering that comes from the upper hemispherical area of the antenna.

Page 13, Line 271: Not all GNSS systems are explicitly designed to reject such signals, but geodetic antennas are.

Page 14, Equations, 10, 11: These need polarization subscripts.

Page 14, Lines 292-293: I completely disagree with this statement. The result is physical as it is measured in the physical world. The reason for negative VOD (or higher under canopy measurements than open-sky ones) can be explained by volume scattering contributions that the antenna is picking up in upper hemispherical region. As stated correctly in the following sentences, this mainly happens at high elevation angles where direct line-of-sight signal go through gaps without much attenuating, but volume scattering within antenna field of view can add additional scattering. This is well aligned with my previous comment on your assumption at Page 8, Line 196.

Page 14, Lines 304-305: As I stated earlier, VOD depends on polarization. Your setup collects data at RHCP-RHCP (RR), but spaceborne measurements are at either H-pol or V-pol. Without physical model justification for the specific trees, the comparison is uncertain.

Page 16, Line 327: It would be good to add a formulate for serial autocorrelation and describe how it is implemented.

Page 19, Line 374: What is the spatial resolution for EVI index?

Page 27, Lines 525-529. I think that these references are misinterpreted. To my knowledge, tree branches scatter and attenuate significantly while leaves attenuate mostly at L-band.

Page 28, Lines 531-533: This assumption is not valid for trees with electrically large branches.

Page 29, Line 577: Again, the model used here is not a physical model. It is a dielectric mixing model.

Page 34, Section 5.3: Rainfall and dew can impact the instrument performance. Without knowledge of instrument under such circumstances, it is difficult to attribute all the changes to the scene.

---

## Referee Comment (RC2)

This reviewer thanks the authors for efforts invested in the preparation of their manuscript. In this paper the authors overview an experimental set up where ground based GNSS receivers are used to passively monitor, primarily, vegetation water content or vegetation optical depth. The paper's topic has been the subject of some investigation in previous works in the relevant literature, but the authors discussion is extensive and offers a number of refreshing views on the topic. This reviewer does note a number of concerns outlined below, but would otherwise recommend accepting the manuscript after revision.

**Comments**

1. Refer to [page 1] "The technique presented here has the potential to resolve two important knowledge gaps, namely the lack of ground truth observations for satellite-based VOD" - This reviewer is a little reluctant to agree with this assertion. While it is true that VOD estimates derived from spaceborne observatories will require some level of ground truth, measurements derived from the proposed sensors will also require ground truth estimates in the calibration of measurements made and validation of subsequent VOD estimates. The authors are encouraged to revise this statement placing the proposed technique in the appropriate context or elaborate to this reviewer on why they feel that the sensors are in situ data, or ground truth data independent.

   It is also important to note that it is difficult for the proposed techniques to really compete with the main impetus for having spaceborne receivers, namely their global coverage versus the proposed highly localized estimates.

2. Refer to [page 2] "Microwave remote sensing methods are broadly categorized as either passive or active. Passive instruments (radiometers)" - The authors are encouraged to revise "Passive instruments (radiometers)" to something along the lines of Passive instruments (like radiometers). Any receiver that does not transmit its own signals or relies on signals transmitted by a none co-located system for sensing is by definition, passive. Radiometers are an example of passive instruments but a wide range of other platforms exist.

3. Refer to [page 3] "Higher VOD values indicate that the canopy is less transparent to microwaves" - The authors are encouraged to generalize this statement to all "impinging or reflected radiation" given that higher VOD also attenuates visible light and IR in larger proportions.

4. Refer to [page 3] "But can hardly be validated, as systematic ground-based VOD observations do not exist at the moment" - It is important to make clear that this is not indicative of an inherent limitation in the ability of spaceborne receivers to provide VOD estimates, just a lack of field campaigns; something that could change in the future and so this reviewer does not regard this as a reasonable example of why the proposed methodology is superior to approaches based on spaceborne receivers' measurements.

5. Refer to [page 9] "(which GNSS antennas are designed to reject)" - The authors are encouraged to make clear that it is ground based GNSS antennas that are designed to receiver RHCP. This is important given that spaceborne GNSS-R receivers typically have antennas that are designed to receive LHCP antennas given GPS signal reflection (and polarization handedness reversal) off the Earth's surface.

6. Refer to [page 11] "from 102 individual GNSS satellites" - This reviewer is only aware of there being 24-32 operational GPS satellites or so, did the authors also use reflections from other GNSS constellations like Galileo, GLONASS and BDS?

7. To make Figure 3(d) a little easier to follow, it may be useful to subject the series to a smooth (mean or median) smoothing filter in 15-1 hour increments. This may also aid in dampening the noise noted on page 14.

Once more this reviewer thanks the authors for efforts invested in the preparation of this manuscript and looks forward to their continued contribution.

---

## Author Response (AR1)

**Contents of the response document**

We reproduce each comment, followed by our response and the corresponding manuscript changes.

Line numbers in comments correspond to the original preprint. Line numbers in our responses correspond to the *revised* manuscript.

In our initial response (author comment: https://doi.org/10.5194/bg-2022-84-AC1), our responses to the third page of comments by referee #1 were accidentally omitted. This has been corrected here.

**1. Response to comments by Referee #1**

This paper uses a pair of two ground-based GNSS receivers to estimate L-band VOD in a forested field. Similar configuration has been previously tested by others. Unique part of this study is for collection of a field data over a period of 8 months with analysis to get insights into vegetation response in different time scales such as hours-minutes for intercepted rainfall and dew, days-hours for canopy water content, and months-weeks for canopy structure. I think that the topic is very important for the community and the approach is very promising to monitor trees characteristics nondestructively in a continuous fashion. However, I have several major concerns on the VOD estimates, the model, and assumptions as they are the basis for the analysis.

> Thank you for the careful evaluation and the constructive review. We provide our point-by-point answers below.

(1) Polarization: The current setup measures the signal in RHCP-RHCP (RR) configuration, which might be significantly different from H-pol or V-pol VOD estimations from the satellites. This relation among various polarizations (RR, H or V pol) has not been established for VOD in the literature, so it is hard to interpret how RR-pol VOD estimates can be used for satellite data validation.

> We agree that the relationship between VOD measured in a RHCP-RHCP configuration and VOD measured from H or V polarizations is poorly known. Existing studies have mostly compared VOD estimated at H and V polarizations and found that $VOD_H$ and $VOD_V$ can have an offset, even though temporal dynamics are in close agreement [Schwank *et al.*, 2021; Guglielmetti *et al.*, 2008; Kurum *et al.*, 2009a]. Our evaluation is that RR-pol VOD estimates would still be useful for comparison with satellite data especially in a relative sense (i.e. evaluation of the temporal dynamics or relative differences across sites). We agree we would strongly advise against validating absolute values at this stage. We add the following text in section 6:

L784: *"In particular, the degree to which GNSS-VOD at RHCP-polarization agrees with other VOD estimates at horizontal (H) or vertical (V) polarization is unknown. For instance, previous studies over forests have shown that H-pol VOD can differ from V-pol VOD, even though temporal dynamics are similar (Schwank et al., 2021;Guglielmetti et al., 2008;Kurum et al., 2009b)."*

(2) Model: The model used here is a dielectric mixing model, which is valid for electrically small constituents. At L-band, primary (even secondary) branches can violate the assumption. In addition, the model ignores the polarization, which could be significant for tree canopies with certain preferred oriented branches.

(3) Assumption: The current model ignores volume scattering. As the data suggests (from my own experience as well), the signals at high elevation under canopy could be larger than the signal in open sky environment. This is a clear indication of volume scattering. The impact of volume scattering may vary from trees to trees and elevation angles to angles. Without physical modeling simulations for various scenarios, it would be difficult to develop methodologies to disentangle the direct signal from volume scattering.

Thank you for these two comments, we fully appreciate these concerns.

A wide range of approaches has been used to relate 1) microwave-range measurements over forests to canopy attenuation, and 2) canopy attenuation to environmental variables of interest like biomass or water content. For instance, some past studies have directly (i.e. empirically) related VOD to vegetation water content or biomass without considering a dielectric mixing model and/or have estimated VOD without taking into account the role of volume scattering [Rodriguez-Alvarez *et al.*, 2012; Vittucci *et al.*, 2019; Brandt *et al.*, 2018]. Other valuable approaches, like the tau-omega model, incorporate more knowledge from microwave radiative transfer theory [Frappart *et al.*, 2020], while still neglecting important aspects like volume scattering, which may be significant in some applications [Kurum *et al.*, 2009b; Feldman *et al.*, 2018].

It is of course possible to make use of more comprehensive formulations, which require making further assumptions regarding the size, density, shape, orientation, and distribution of the larger canopy elements. Using such more complex formulations, *Guerriero et al. [2020]* concluded that, for an experimental setup comparable to ours, direct signals largely dominate over volume scattering. In addition, they show that volume scattering produces signal fluctuations with timescale in the order of a few seconds that tend to average out with time integration (as the transmitting satellite moves through the sky). Here we analyse hourly VOD time series based on measurements averaged over the whole hemisphere such that these effects are not dominant in our opinion (although we fully agree that they could be important if we were to, say, analyse instantaneous values or long-term averages taken at a very localized azimuth and elevation). *Schwank et al. [2021]* analyse data from an upward-looking radiometer and also argue that volume scattering within the canopy plays a minor role at L-band and neglect it in their model of

VOD. Obviously, this does not mean that scattering effects would be minor in all configurations, for instance, they are likely important to consider in a reflectometry (GNSS-R) experiment or when analysing below-canopy LHCP signals.

Using the MIMICS model, *Steele-Dunne et al. [2012]* concluded that leaf moisture is by far the dominant control on forest transmissivity at L-band for H and V polarizations (though branches and trunks do contribute significantly to backscatter). Thus, we believe using a dielectric mixing model valid for electrically small constituents such as leaves is acceptable, even though we agree with the reviewer that this assumption may be violated for larger elements.

In this study, our goal is primarily to showcase the potential of GNSS-based remote sensing for deriving useful information on vegetation biomass and water content. We document and discuss the limitations of our modelling approach at several points in the manuscript and aim at making all assumptions 100% transparent. Given the scope of the journal, the objectives of our study, the findings of both *Guerriero et al. [2020]* and *Steele-Dunne et al. [2012]*, and the available data, we believe our approach to be a reasonable compromise in terms of model complexity. Future work could certainly explore more complex formulations.

In response to the referee's concerns, we add these further statements to the manuscript.

*(concerning the dielectric mixing model, L202): "While this may be true for leaves, this assumption may not hold for larger elements such as branches and trunks."*

*(later in the Methods section, L232): It is important to note that this simple formulation neglects several aspects, including volume scattering, which may be important in configurations with denser biomass (and also when interpreting RHCP backscatter or LHCP signals).*

*(at the end of the following sentence of the conclusion, L745)*: In a further step, we use existing *simple* models of canopy microwave transmissivity and vegetation dielectric parameters [Ulaby and Long, 2014] to demonstrate the feasibility of using such GNSS-based VOD observations to derive information on canopy water content and aboveground dry biomass *(future work could certainly explore more complex formulations)*.

Finally, we would like to stress that observing slightly higher GNSS power at the subcanopy site compared to the open-sky reference does not necessarily indicate significant volume scattering as implied by the reviewer. Even when placing two identical instruments within 2 meters of each other in an open-sky environment, small differences in measured signal power will unavoidably occur between the instruments (see below, data from an earlier test case with two Trimble NetRS instruments). These differences may arise from random noise, as well as non-random minor differences in antenna gain patterns, ground multipath effects, etc. Note that while these differences contribute some

non-random noise in a hemispheric representation like the one shown in Figure 4a, they largely cancel out in our analysis of the hourly and hemispheric-integrated average.

[Figure]

Supporting Figure 1. Raw data collected by two GNSS receivers located next to each other at a site with open-sky conditions. While there is no bias between the instruments on average, small deviations always occur due to measurement errors (also see L339-344 of the manuscript).

Additional references (use a similar approach) to consider:

https://doi.org/10.1109/IGARSS47720.2021.9555155
https://doi.org/10.1155/2017/6941739

Thank you very much for making us aware of these references, we have integrated them.

L127: *"Kurum and Farhad (2021) also tested it with mobile GNSS antennas and Zribi et al. (2017) used it to monitor sunflower canopies."*

Specific Comments:

Page 1, Line 16: I think you meant "signal strength".

Thank you, corrected.

Page 1, Line 24: Have you checked the impact of rainfall and dew on the instrument? Any water on the antenna surface can change its radiation characteristics.

That's a very good point! Yes, we have looked at this question using the signal strengths of the 3 geostationary satellites of the SBAS/WAAS system which are also tracked by our receivers. Since these satellites have a fixed position in the sky their signal strength is usually quite stable, which makes them ideal for looking at potential effects of water on the surface of the antenna. We found that the signal strength at the clear-sky site is unaffected by the rain event, even though the nearby surface wetness sensor was saturated. This suggests that water on the antenna surface is not a primary concern for our type of antenna.

[Figure]

*Supplementary Figure 2. Signal strengths of the three geostationary SBAS/WAAS satellites measured at the reference and the forested sites during the rainfall event on May 18, 2020. The signal strength of the reference antenna remains stable, suggesting that the presence of water on the antenna surface does not cause strong perturbations (for our type of antenna). Note that the signal is attenuated differently for each satellite at the forested site because of the heterogeneous canopy distribution.*

We will integrate this figure to the supplementary information and mention it in the main text.

L689: *"We could also establish that signal strength was likely not affected by the presence of intercepted water on the antenna itself (Supplementary Figure S2)."*

Page 2, Line 29: I think you meant "receiver systems".

Thank you, corrected.

Page 2, Line 40: I would say "continuously gathered network of ground truth data".

Changed to: *"a network of continuously gathered ground truth data does not exist yet"*

Page 2, Line 53: Replace "either or" with "both and" as both mentioned factors impact the transmissivity simultaneously.

Thanks for spotting this, corrected.

Page 3, Equation (1) : Add polarization symbol to this equation since VOD and gamma depends on polarization.

That's absolutely right, however we'd prefer to keep the generic version of that equation (i.e. as in *Frappart et al. [2020]* and many others). We modify the following statement instead:

*L62:* "where VOD represents an attenuation coefficient (specific to the observation wavelength *and polarization*)"

Page 4, Footnote: Coarse spatial resolution is not due to low signal energy. It is due to incoherent nature of the signal that requires real aperture antenna (limited in size).

Thanks, we aimed to span over a wide range of methods (from spaceborne radiometers to SAR). We clarified the sentence as follows: *"Drawbacks include a lower energy and longer wavelength, which usually translates into coarser spatial and/or radiometric resolution".*

Page 5, Line 114: Please state the polarization of antennas used with these receivers here. In addition, receivers could be placed on the ground (without tripod) as long as they are levelled and cleared from the surrounding obstructions.

You're right. In our case, placing them on the ground was not possible. We now mention the polarization at L159:

*"At each site a Septentrio PolaRx5e GNSS receiver, connected to a PolaNt-x MF (RHCP) GNSS antenna, measured multi-constellation GNSS signals over the period of May 13 to December 10, 2020, with a logging rate of 15 seconds."*

And the positioning at L774:

*"It can even be placed on the ground (without a tripod) as long as it is level and free from obstructions."*

Page 7, Lines 149 - 151: The pictures of receiver set-up would be useful to add to the manuscript.

Good idea. We add the following images as Supplementary Figure.

[Figure]

*Supplementary Figure S3. (a) Antennas under test at the open-sky site (the PolaNt-x MF antennas are the smaller ones at the back). (b) Receivers. (c) Deployment at the forested site. The other larger Zephyr antenna (on the yellow tripod) was tested as a backup solution but was not used in the end.*

Page 7, Line 156: Please state the polarization of the antenna.

Done.

Page 8, Line 179: This model is "dielectric mixing model". The physical model implies either "discrete scatter" based radiative transfer or wave theory, none of which are used in this paper. I would suggest not to use term "physical" for the model.

Thank you for this suggestion. We replace "physical model" with "dielectric mixing model".

Page 8, Line 181: This is a major assumption which is not true for well-developed trees. Tree branches scatter and attenuate significantly while leaves attenuate mostly at L-band, so the received signal is expected to be a blend of both volume scattering and attenuation, depending on the vegetation, polarization, and incidence angle.

Yes, this assumption is discussed a few lines further at L202-207 as well as L550-560 and L594-597. Also see our response to the main comments discussing the literature supporting this assumption for our case and the statements we added to make these assumptions 100% transparent throughout the manuscript.

Page 8, Equation (4): This is repetition of Equation (1). I would define VOD here, instead.

We understand this comment. Our intention is for equation (1) to remain relatively generic as it appears in the introduction. In the methods, equation (4) provides one formulation (one among many possible) based on the assumed dielectric mixing model. We'd like to keep these separate.

Page 8, Equation (5): This formula is applicable if inclusions are much smaller than wavelength as you correctly stated in the following paragraph. The formula can be applied to randomly oriented agricultural crops with no preferred scattering orientation and smaller than wavelength, but the validity region would be way off for tree canopies. The more descriptive formulation can be written as a function of forward scattering amplitudes of individual tree constituents. An example can be found at equation (8) of reference Guerriero et al. 2020.

We now mention this limitation explicitly at L202 (also see our response to the main comment for a discussion).

*"While this may be true for leaves, this assumption may not hold for larger elements such as branches and trunks."*

Page 8, Line 196: That is correct that Guerriero et al. 2020 showed that coherent line-of-sight signal mostly dominates over volume scattering for RHCP-transmit RHPC-receive cases. However, it is hard to generalize this statement as this behavior can depend on the elevation, biomass, and type of trees. The question is if it is applicable to your case.

Our setup seems comparable to that of Guerriero et al. 2020 (their figure 3), with perhaps generally more randomly distributed canopy elements in our case. We agree that a more comprehensive investigation of the potential contribution of volume scattering to RHCP-transmit RHCP-receive cases in a wide variety of situations would be very welcome. Hopefully this can be the subject of future work. In the meantime, we now discuss this more explicitly at L232:

*"It is important to note that this simple formulation neglects several aspects, including volume scattering, which may be important in configurations with denser biomass (and also when interpreting RHCP backscatter or LHCP signals)"*

Page 9, Lines 231-214: This semi-empirical formula is developed for agricultural crops. It is not clear how it can be applied to trees, which are electrically much larger at L-band.

This was investigated by *Chuah et al. [1995]*. They compared 3 different semi-empirical dielectric models against measurements made on leaves from rubber trees and oil palms. They conclude that the dual-dispersion model of Ulaby and El-Rayes provides good

estimates in most cases. The same formula was used for forests in *Steele-Dunne et al. [2012]* for example.

Page 9, Line 220: I would call the dielectric mixing model approach as "zero-order" as the first order has completely different meaning in physical models such as RTE or wave theory.

We replace "first-order" with "simple" to avoid any misinterpretation.

Page 11, Lines 248-249: I would say "yield information on VOD" as stated at page 5, Line 116 as I stated above difference SNR can include some of the volume scattering.

We replace "mainly" by "for the most part". Also see our response to the main comments.

Page 12, Lines 260-261, Page 13, Lines 275-276: It is true that ground reflections would be small since your antenna is designed to reject reflected LHCP signals from the ground, but it would still receive volume scattering that comes from the upper hemispherical area of the antenna.

We modify the sentence at L281-284 as follows:

*It should be noted that while the SNR measurement is dominated by the contribution of the direct (line of sight) signal, it also includes a comparatively weaker contribution from volume scattering and indirect multipath reflections (Bilich et al., 2007;Smyrnaios et al., 2013), the latter of which may contain information on soil water status for instance (Larson, 2016).*

Page 13, Line 271: Not all GNSS systems are explicitly designed to reject such signals, but geodetic antennas are.

This is an important point. We modify the text as follows:

L292: … *"even though most geodetic ground-based GNSS systems are explicitly designed to reject such signals"*

L295: *"(note that in contrast, spaceborne GNSS reflectometry also relies on the LHCP signal)."*

Page 14, Equations, 10, 11: These need polarization subscripts.

We add the sentence:

*"Note that in our case, this represents L-band VOD at RHCP polarization."*

Page 14, Lines 292-293: I completely disagree with this statement. The result is physical as it is measured in the physical world. The reason for negative VOD (or higher under canopy measurements than open-sky ones) can be explained by volume scattering contributions that the antenna is picking up in upper hemispherical region. As stated correctly in the following sentences, this mainly happens at high elevation angles where direct line-of-sight signal go through gaps without much attenuating, but volume scattering within antenna field of view can add additional scattering. This is well aligned with my previous comment on your assumption at Page 8, Line 196.

Thank you for your comment. As shown in our response to the main comment, negative instantaneous VOD measurements also simply occur because of measurement errors. We demonstrate this in our response by comparing two antennas located at a location with no canopy. Thus, observing some negative VOD in canopy gaps in our case does not demonstrate that volume scattering would be more significant than what is suggested in *Guerriero et al. [2020]*.

Page 14, Lines 304-305: As I stated earlier, VOD depends on polarization. Your setup collects data at RHCP-RHCP (RR), but spaceborne measurements are at either H-pol or V-pol. Without physical model justification for the specific trees, the comparison is uncertain.

Thank you for the comment. We modify the sentence as follows:

"*The whole canopy average VOD is 0.79, which is similar to what is reported for evergreen broadleaf forests at L-band (Konings et al., 2017a).*"

Note that while Konings et al. 2017 use both H-pol and V-pol in their retrieval algorithm, they assume that vegetation parameters (including VOD) are not polarization-dependent for the sake of parsimony, consistent with other passive microwave retrievals.

We added the following statement in the conclusion as well:

L784: "*In particular, the degree to which GNSS-VOD at RHCP-polarization agrees with other VOD estimates at horizontal (H) or vertical (V) polarization is unknown. For instance, previous studies over forests have shown that H-pol VOD can differ from V-pol VOD, even though temporal dynamics are similar (Schwank et al., 2021;Guglielmetti et al., 2008;Kurum et al., 2009b).*"

Page 16, Line 327: It would be good to add a formulate for serial autocorrelation and describe how it is implemented.

We add the following footnote.

*"The serial autocorrelation is calculated using the time series shown in Fig. 7a-b and is defined here as the Pearson correlation coefficient between the time series at time t and t-l, where l is the lag."*

Page 19, Line 374: What is the spatial resolution for EVI index?

We added the resolution in this sentence:

"Using quality-checked *30-meter* satellite images from Sentinel 2, …"

Page 27, Lines 525-529. I think that these references are misinterpreted. To my knowledge, tree branches scatter and attenuate significantly while leaves attenuate mostly at L-band.

Together these references suggest that while tree branches do scatter and attenuate significantly [Kurum *et al.*, 2009a], this signal dominates the received signals as a whole mostly when measuring backscatter or when measuring the one-way attenuation of a canopy without leaves. When measuring the one-way attenuation of a canopy with full leaves, attenuation due to leaves dominates [Steele-Dunne *et al.*, 2012; Guerriero *et al.*, 2020].

Page 28, Lines 531-533: This assumption is not valid for trees with electrically large branches.

Thank you for this comment. Yes, this paragraph is meant to discuss the limitations of this assumption.

We also add the following comment in the discussion.

*"Note that this estimate should be interpreted with the awareness that VOD-based estimates of AGB likely do not weigh all canopy constituents evenly. While L-band VOD is primarily sensitive to leaves, the sensitivity to branches and trunks can also increase at lower leaf moisture content [Steele-Dunne et al., 2012]."*

Page 29, Line 577: Again, the model used here is not a physical model. It is a dielectric mixing model.

We replace *"simple but physics-driven model"* with *"simple model"*

Page 34, Section 5.3: Rainfall and dew can impact the instrument performance. Without knowledge of instrument under such circumstances, it is difficult to attribute all the changes to the scene.

Thank you, we have addressed this question in our above response to the reviewer's comment (about Page 1, Line 24).

We thank the reviewer for their time and their constructive feedback!

**2. Response to comments by Referee #2**

This reviewer thanks the authors for efforts invested in the preparation of their manuscript. In this paper the authors overview an experimental set up where ground based GNSS receivers are used to passively monitor, primarily, vegetation water content or vegetation optical depth. The paper's topic has been the subject of some investigation in previous works in the relevant literature, but the authors discussion is extensive and offers a number of refreshing views on the topic. This reviewer does note a number of concerns outlined below, but would otherwise recommend accepting the manuscript after revision.

Thank you for the careful evaluation and the constructive review. We provide our point-by-point answers below.

Comments

1. Refer to [page 1] "The technique presented here has the potential to resolve two important knowledge gaps, namely the lack of ground truth observations for satellite-based VOD" - This reviewer is a little reluctant to agree with this assertion. While it is true that VOD estimates derived from spaceborne observatories will require some level of ground truth, measurements derived from the proposed sensors will also require ground truth estimates in the calibration of measurements made and validation of subsequent VOD estimates. The authors are encouraged to revise this statement placing the proposed technique in the appropriate context or elaborate to this reviewer on why they feel that the sensors are in situ data, or ground truth data independent. It is also important to note that it is difficult for the proposed techniques to really compete with the main impetus for having spaceborne receivers, namely their global coverage versus the proposed highly localized estimates.

Thank you for this comment. Our intention is not to compete with satellite microwave sensors, in fact it is primarily to support their development that we originally initiated this study. We fully agree with the reviewer that the proposed technique cannot compete with spaceborne receivers in terms of their global coverage and global relevance.

The comment on GNSS-based VOD being or not being ground truth is an interesting point. Maybe it is helpful to think of in situ soil moisture measurements, as they are today widely accepted as ground truth for satellite-based retrievals. Data from the International Soil Moisture Network is used extensively as ground truth by most centres. This data mainly relies on TDR probes which also need to be calibrated and evaluated against

further measurements, for instance from gravimetric soil samples. Given this context, we do believe that GNSS-based VOD has in fact the potential to provide independent ground truth observations. This is not to say that one should expect GNSS-based VOD to match one-to-one with satellite retrievals. As the reviewer points out, we do not know yet how well GNSS-based VOD compares against other estimates of VOD. We now mention this more explicitly (see below).

L784: *"In particular, the degree to which GNSS-VOD at RHCP-polarization agrees with other VOD estimates at horizontal (H) or vertical (V) polarization is unknown. For instance, previous studies over forests have shown that H-pol VOD can differ from V-pol VOD, even though temporal dynamics are similar (Schwank et al., 2021;Guglielmetti et al., 2008;Kurum et al., 2009b)."*

The difference in scale and footprint between satellite and ground-based data also constitutes an important obstacle. This is already mentioned at L799: *"For example, arrays of GNSS receivers deployed within the spatial footprint of a satellite VOD grid cell (i.e. about 30 km) may serve to estimate a regional average VOD that would be suitable as ground truth for the satellite products."*

2. Refer to [page 2] "Microwave remote sensing methods are broadly categorized as either passive or active. Passive instruments (radiometers)" - The authors are encouraged to revise "Passive instruments (radiometers)" to something along the lines of Passive instruments (like radiometers). Any receiver that does not transmit its own signals or relies on signals transmitted by a none co-located system for sensing is by definition, passive. Radiometers are an example of passive instruments but a wide range of other platforms exist.

Thank you for pointing this out. We have made the suggested change.

3. Refer to [page 3] "Higher VOD values indicate that the canopy is less transparent to microwaves" - The authors are encouraged to generalize this statement to all "impinging or reflected radiation" given that higher VOD also attenuates visible light and IR in larger proportions.

Agreed. We replace with: *"Higher VOD values indicate that the canopy is less transparent to electromagnetic waves."*

4. Refer to [page 3] "But can hardly be validated, as systematic ground-based VOD observations do not exist at the moment" - It is important to make clear that this is not indicative of an inherent limitation in the ability of spaceborne receivers to provide VOD estimates, just a lack of field campaigns; something that could change in the future and so this reviewer does not regard this as a reasonable example of why the proposed methodology is superior to approaches based on spaceborne receivers' measurements.

Thank you for this comment, in fact, we fully agree with the reviewer's statement. We do not think the text argues that the proposed methodology is superior to spaceborne receivers. See for instance some of our current statements:

*L42: "We then present a ground-based technique relying on Global Navigation Satellite Systems (GNSS) with the objective to address the lack of ground-based VOD observations."*

*L100: "Considering some advantages of microwave-range compared to visible-range observations, such studies have demonstrated the interest of VOD for monitoring vegetation dynamics from space (Konings et al., 2021)."*

*L797: "The results presented here suggest that GNSS-based VOD may have the potential to fill a key research gap in terms of linking satellite-based L-band VOD observations to ground observations."*

5. Refer to [page 9] "(which GNSS antennas are designed to reject)" - The authors are encouraged to make clear that it is ground based GNSS antennas that are designed to receiver RHCP. This is important given that spaceborne GNSS-R receivers typically have antennas that are designed to receive LHCP antennas given GPS signal reflection (and polarization handedness reversal) off the Earth's surface.

This is an important point. We have made this clearer:

L206: *"(which most geodetic ground-based GNSS antennas are designed to reject)"*

L295: *"(note that in contrast, spaceborne GNSS reflectometry also relies on the LHCP signal)."*

6. Refer to [page 11] "from 102 individual GNSS satellites" - This reviewer is only aware of there being 24-32 operational GPS satellites or so, did the authors also use reflections from other GNSS constellations like Galileo, GLONASS and BDS?

Indeed, this is mentioned at L358 and L735. We now also mention it in the methods:

L159: *"At each site a Septentrio PolaRx5e GNSS receiver, connected to a PolaNt-x MF (RHCP) GNSS antenna, measured multi-constellation (GPS, GLONASS, Galileo, BeiDou) GNSS signals ..."*

7. To make Figure 3(d) a little easier to follow, it may be useful to subject the series to a smooth (mean or median) smoothing filter in 15-1 hour increments. This may also aid in dampening the noise noted on page 14.

Yes, this is inherently what is done later in the analysis when hourly VOD time series are calculated. However, we would like Figure 3d to show the raw data and thus support the discussion on page 14.

Once more this reviewer thanks the authors for efforts invested in the preparation of this manuscript and looks forward to their continued contribution.

We thank the reviewer for their time and their constructive feedback!

**3. Response to comments by Referee #3**

The authors present an experimental technique using GNSS that can be used to provide continuous VOD estimates. The paper is well-written and the level of detail provided is generally appropriate. Given that the paper is likely to be read by a wide audience, many of whom are not familiar with GNSS, I would recommend including some additional details (see below). The technique and methodology presented have the potential to be hugely valuable for the microwave remote sensing community, particularly those concerned with observing soil and vegetation. The authors provided valuable recommendations on the deployment of similar set-ups, and outline several potential applications. This paper is highly innovative, timely and can be expected to have a significant impact in the field of hydrology and remote sensing. To my knowledge, the theory and methodology are sound. I recommend that it is accepted for publication in this special issue if the following comments can be addressed.

Thank you for the careful evaluation and the constructive review. We provide our point-by-point answers below.

Major comments:

In lines 554-556, and the discussion in lines 569-589, AGB and CWC are used to refer to the total aboveground portion of the vegetation, including leaves, branches, trunks etc.. However, as discussed in lines 523-531, observations and modeling studies have shown that L-band transmissivity is primarily sensitive to leaves. This suggests that the GNSS VOD produced here is primarily sensitive to leaves and that the dynamics observed in GNSS VOD are primarily due to variations in leaves with the sensitivity to branches and trunks depending on leaf moisture content. The same is also true for other L-band VOD products. Nonetheless, I think this should be mentioned in lines 565-576 as it provides some explanation for the difference among the estimates based on the models of Vitucci and Brandt. It also serves as a caution to users on the interpretation of AGB derived from VOD.. It is also relevant for the discussion of CWC because (1) the CWC is calculated using the estimated AGB and (2) the definition of "canopy", in the sense of which constituents are observed, varies depending on leaf moisture content - the dynamics in this CWC are expected to be primarily due to leaf water dynamics.

Thank you for this comment, this is a very good point. We add the following statements:

L595: *"Note that this estimate should be interpreted with the awareness that VOD-based estimates of AGB likely do not weigh all canopy constituents evenly. While L-band VOD is primarily sensitive to leaves, the sensitivity to branches and trunks can also increase at lower leaf moisture content (Steele-Dunne et al., 2012)."*

L608: *"As for AGB, it's important to keep in mind that the CWC estimate does not weigh all canopy constituents evenly"*

Define what is meant by canopy in the paper. Is it used to mean the aboveground portion of the vegetation? The portion above the sensor? Or the upper layer of the forest? It is important to be clear here because the paper is likely to be read from both the remote sensing community as well as the forest ecology community. This is also relevant in the context of the discussion above regarding canopy water content.

Thank you for raising this point. We now define canopy explicitly:

L116: *"Here, canopy is understood as the portion of vegetation lying above the sensor (in our case, this excludes the forest floor and ground vegetation)."*

Lines 107-110: I would not use "proxy" here. GNSS-VOD should not be considered a direct proxy for biomass or leaf water status. The current formulation suggest that the relationship between GNSS-VOD and biomass and leaf water status is more direct than it is. It is fine to say that GNSS-VOD could be useful to interpolate and gap-fill sparse and labour-intensive measurements of biomass and leaf water status but there many assumptions and models needed between the two.

We fully agree with the reviewer that there is a long way from GNSS-VOD to these quantities. We believe the term of "proxy" is still warranted, considering the current practice in other studies on VOD, for instance in the same journal [e.g. Mucia *et al.*, 2022; Schmidt *et al.*, 2022].

At L109: we replace "… proxy …" with "… indirect proxy …"

Section 3.1: For readers not familiar with GNSS, it would be helpful here to provide some description of how the data shown in Figure 3 are obtained in terms of satellite overpasses, viewing geometry etc.. A short description of GNSS constellations would be helpful. It would also be helpful to explain how Figure 3 should be read in terms of azimuth and incidence/elevation angle. Please label azimuth and incidence angle on the plots and/or mention in the caption to improve readability for new users.

That's a good idea. We have updated Figure 3 and its caption to make it more readable for readers not familiar with GNSS (see below).

[Figure]

*Figure 3. Sky plots illustrating the SNR observations on May 21, 2020 for one specific GPS satellite (PRN2) at the open sky site (a) and the forested site (b). (c) Difference in SNR between the two sites. (d) Same as a-c but showing the temporal evolution of the SNR. The centre of the polar plots corresponds to the local zenith.*

The text is also modified to include the following description:

L254: *"The hemispherical plot in Fig. 3a illustrates the SNR values measured over the course of one single day at the reference (open-sky) station for just one satellite of the GPS constellation (PRN2). GNSS satellites are commonly identified by their pseudorandom code (PRN) which allows the receiver to determine which satellite is being tracked, such that its azimuth and elevation can be calculated. Individual satellite tracks repeat after a period that depends on the GNSS constellation (e.g. twice per sidereal day (23h56) for GPS and every 10 sidereal days for Galileo)."*

Section 3.1: Provide details on how data from different GNSS constellations are merged. In particular, mention if there are any systematic differences and how they are handled during merging.

Thanks for this suggestion. We add the following:

*L272: "Note that absolute SNR values vary from spacecraft to spacecraft, as those have different (and occasionally time-varying) transmit powers. It is thus very important to first pair the individual SNR measurements taken by the two receivers and only then, average the obtained ΔSNR values."*

Line 294 – 302: I'm not convinced by this argument. If non-random multipath interferences are not excluded, will they not introduce or contribute to spurious values of VOD rather than random noise? If so, there is a danger that these are incorrectly interpreted as VOD variations? Please demonstrate that this is not the case.

Thank you for this comment. This statement refers only to temporal averages but we agree the formulation was maybe a bit unclear.

As the non-random coherent reflected multipath signals go in and out of phase, this produces peaks and lows in the collected SNR data [Nievinski and Larson, 2014]. When comparing data from two different receivers, these peaks and lows are not aligned, potentially leading to values where SNR at the reference site is lower than at the forested site. Our argument is that these differences average out with time integration (the same argument is made in *Guerriero et al. [2020]*). See the following figure based on data collected with two antennas placed next to each other at an open-sky site. In this case, there is no vegetation so our VOD estimate should be zero. If we were to exclude negative VOD values and only average positive VOD values, our hourly average VOD would be biased high. We make it clearer in the text that this applies to temporal averages.

The following statement…

L320: *"To preserve the error structure of the measurements, we propose to still use these unphysical values whenever possible, and especially when computing averages, so that positive and negative random errors can cancel out, avoiding a potential bias in our estimate of the long-term average VOD."*

is modified as follows:

*"To preserve the error structure of the measurements, we propose to still use these unphysical values when computing temporal (i.e. daily or hourly) averages later in the paper, so that positive and negative errors can cancel out, avoiding a potential bias in our estimate of the average VOD."*

[Figure]

Top: Open-sky SNR values measured with two Trimble NetRS receivers placed next to each other and equipped with Zephyr antennas on November 30, 2019, for a satellite of the GPS constellation (PRN2). Bottom: Histogram of all individual VOD values derived from all satellites over a period of 1 hour (and averages of that data).

Section 3.3: I found the nomenclature in this section, particularly the use of the terms anomaly and static, confusing and potentially misleading. In practice, the issue is that a robust estimate of the temporal variation can only be obtained at the expense of spatial aggregation, i.e. a loss of spatial resolution. The methodology to obtain the time series itself is fine, but I would recommend re-thinking the nomenclature.

Thank you for sharing these concerns. We reformulate the nomenclature as follows.

*L364: "The goal is to subtract the angular heterogeneity in VOD, representing the uneven canopy distribution, and only retain residuals from the locally averaged attenuation (Eq. 12). The long-term average at a given incidence angle and azimuth (Eq. 13) is calculated inside a neighbourhood N that includes all measurements within some chosen angular distance δ from that point of interest (Eq. 14)."*

And add the following:

*L343: "In practice, this means that a continuous (gap-free) and robust VOD time series can only be obtained by aggregating data collected at different azimuth and elevation angles (i.e. trading angular resolution for temporal coverage)."*

Lines 495 to 507: Why is it necessary to optimize v_veg with a daily time step? In a forest, in particular, this quantity is likely to vary over much longer time scales. This could obviate the need for some of the low-pass filtering in later steps.

That's a good suggestion, in fact, calibrating v_veg at a longer time scale is what we did initially. However, optimizing v_veg at a daily time step provides a more efficient way of mitigating the influence of outliers (like the rainfall event at the beginning of the time series). With a daily estimate, only the v_veg of a single day is heavily biased and this is easily removed with the low-pass filter. If we were to optimize v_veg over a moving period of say, a whole week, the whole week may be biased high around that event.

The conclusion should include some discussion of the trade-off between temporal and spatial resolution. Lines 460-466 could be moved to the conclusion as part of this discussion. It is relevant in terms of the processing, but also in terms of sensor installation. I think it is important to emphasize that the capacity to obtain finer angular resolution comes at the expense of temporal resolution. There are applications where one might be more critical than the other, and many applications where the trade-off is non-trivial.

We agree. We add more discussion on this in the conclusion:

L738: *"Here, obtaining such high-frequency (e.g. hourly) VOD time series comes at the cost of angular resolution, since measurements taken at all azimuths and elevation angles are aggregated into hourly averages. Because of the configuration of the GNSS orbits, users face a trade-off between obtaining VOD estimates at high angular resolution (e.g. Figure 5b) versus obtaining VOD time series at high temporal resolution (e.g. Figure 7b)."*

Minor comments:

I would recommend having it proof-read by a native speaker to remove small errors.

We will do this. In the meantime, thank you for your corrections below.

Line 11: time-consuming destructive samples

Line 18: at a forested site

Line 24: Sensitivity to rainfall and dew deposition events ….

Line 33: remove "direct". The information is not direct. It needs to be inferred from retrieval products.

Thanks! All done.

Line 39: re-phrase. The use of arguably and currently is awkward.

We remove these two words to make the sentence simpler.

Line 166: How many leaf samples? Provide details of the protocol used to ensure that the leaves collected were representative.

Here is the updated description:

L170: *"Forty-eight leaf samples were collected from two live oaks closest to the GNSS antenna on October 18, 2020, at 7am, 12pm and 5pm using a 2m long pruner. For each tree we equally sampled the same three different parts of the crown. Unless otherwise stated below, we followed the protocol advised in Mullan and Pietragalla [2012]. Leaves were weighed on-site immediately after being sampled (fresh weight; FW) and stored individually in cooled glass vials."*

Line 228: Define vegetation density for readers not familiar with microwave remote sensing.

We update the definition at L214: *"where v_veg represents the vegetation volumetric density, defined as the volume fraction of the vegetation material within the canopy (on the order of 0.0001-0.01 m3/m3), a parameter that may vary as a function of the growth cycle"*

And we add the following:

*"This parameter is not to be confused with other measures of vegetation density like crown volume (i.e. including empty space) per m² for instance."*

Figure 2: In the caption, replace "Canopy transmissivity" with GNSS VOD.

We replace with "Modelled VOD".

Line 247: This should be Eq.9 ?

Yes, thanks a lot for spotting this.

Line 279: It would be useful to indicate which data are excluded on Figure 3(d), in terms of time of day so that the reader can put the discussion in this section in the context of the data they see in Figure 3.

Done, we have added the incidence angles in the revised Figure 3d (see above).

Line 419: The study of Vermunt describes a diurnal cycle in backscattern (not VOD) due to dew and interception. It belongs in the first paragraph of this section.

That's right. We reformulate L444: *"Overall, our results agree with previous observations of a diurnal cycle in VOD and backscatter (e.g. Konings et al., 2017b;Holtzman et al., 2021;Vermunt et al., 2021;Prigent et al., 2022)."*

Line 454: v_veg is called the volume density here and the vegetation density elsewhere. Define it once, clearly, in Line 228 and use a single term throughout.

We have made sure we use "vegetation volumetric density" everywhere.

Line 506: What metrics are used to evaluate agreement?

We made our statement clearer:

L514: *"The root mean square error between modelled and observed VOD is always used as the cost function and optimization at steps #2 and #4 is carried with a simplex search method."*

Line 515: What is the cost function used here?

See above.

Line 531 – 533: Remove "Thus," from this sentence. The assumption does not follow from the previous two sentences. Though it is a necessary assumption. You should write "It is assumed that the dielectric … ".

Agreed.

Line 538: the retrieved gravimetric …

Line 715: time of the Sentinel-2 overpass

Line 739: We suggest placing …

Thanks, we have made the suggested changes.

We thank the reviewer for their time and their constructive feedback!

Brandt, M., J.-P. Wigneron, J. Chave, T. Tagesson, J. Penuelas, P. Ciais, K. Rasmussen, F. Tian, C. Mbow, A. Al-Yaari, N. Rodriguez-Fernandez, G. Schurgers, W. Zhang, J. Chang, Y. Kerr, A. Verger, C. Tucker, A. Mialon, L. V. Rasmussen, L. Fan, and R. Fensholt (2018), Satellite passive microwaves reveal recent climate-induced carbon losses in African drylands, *Nature Ecology & Evolution*, *2*(5), 827-835.

Chuah, H. T., K. Y. Lee, and T. W. Lau (1995), Dielectric constants of rubber and oil palm leaf samples at X-band, *Ieee T Geosci Remote*, *33*(1), 221-223.

Feldman, A. F., R. Akbar, and D. Entekhabi (2018), Characterization of higher-order scattering from vegetation with SMAP measurements, *Remote Sensing of Environment*, *219*, 324-338.

Frappart, F., J.-P. Wigneron, X. Li, X. Liu, A. Al-Yaari, L. Fan, M. Wang, C. Moisy, E. Le Masson, Z. Aoulad Lafkih, C. Vallé, B. Ygorra, and N. Baghdadi (2020), Global Monitoring of the Vegetation Dynamics from the Vegetation Optical Depth (VOD): A Review, *Remote Sensing*, *12*(18).

Guerriero, L., F. Martin, A. Mollfulleda, S. Paloscia, N. Pierdicca, E. Santi, and N. Floury (2020), Ground-Based Remote Sensing of Forests Exploiting GNSS Signals, *Ieee T Geosci Remote*, *58*(10), 6844-6860.

Guglielmetti, M., M. Schwank, C. Matzler, C. Oberdorster, J. Vanderborght, and H. Fluhler (2008), FOSMEX: Forest Soil Moisture Experiments With Microwave Radiometry, *Ieee T Geosci Remote*, *46*(3), 727-735.

Kurum, M., R. H. Lang, P. E. O'Neill, A. T. Joseph, T. J. Jackson, and M. H. Cosh (2009a), L-Band Radar Estimation of Forest Attenuation for Active/Passive Soil Moisture Inversion, *Ieee T Geosci Remote*, *47*(9), 3026-3040.

Kurum, M., R. H. Lang, C. Utku, and P. E. O'Neill (2009b), A physical model for microwave radiometry of forest canopies, in *2009 IEEE International Geoscience and Remote Sensing Symposium*, edited, pp. III-294-III-297.

Mucia, A., B. Bonan, C. Albergel, Y. Zheng, and J.-C. Calvet (2022), Assimilation of passive microwave vegetation optical depth in LDAS-Monde: a case study over the continental USA, *Biogeosciences*, *19*(10), 2557-2581.

Mullan, D., and J. Pietragalla (2012), Chapter 5. Leaf relative water content, in *Physiological breeding II: a field guide to wheat phenotyping*, edited by A. Pask, J. Pietragalla, D. Mullan and M. P. Reynolds, p. 132, CIMMYT, Mexico.

Nievinski, F. G., and K. M. Larson (2014), Inverse Modeling of GPS Multipath for Snow Depth Estimation—Part I: Formulation and Simulations, *Ieee T Geosci Remote*, *52*(10), 6555-6563.

Rodriguez-Alvarez, N., X. Bosch-Lluis, A. Camps, I. Ramos-Perez, E. Valencia, H. Park, and M. Vall-llossera (2012), Vegetation Water Content Estimation Using GNSS Measurements, *IEEE Geoscience and Remote Sensing Letters*, *9*(2), 282-286.

Schmidt, L., M. Forkel, R.-M. Zotta, S. Scherrer, W. A. Dorigo, A. Kuhn-Régnier, R. van der Schalie, and M. Yebra (2022), Assessing the sensitivity of multi-frequency passive microwave vegetation optical depth to vegetation properties *Biogeosciences Discussions*.

Schwank, M., A. Kontu, A. Mialon, R. Naderpour, D. Houtz, J. Lemmetyinen, K. Rautiainen, Q. Li, P. Richaume, Y. Kerr, and C. Mätzler (2021), Temperature effects on L-band vegetation optical depth of a boreal forest, *Remote Sensing of Environment*, *263*.

Steele-Dunne, S. C., J. Friesen, and N. van de Giesen (2012), Using Diurnal Variation in Backscatter to Detect Vegetation Water Stress, *Ieee T Geosci Remote*, *50*(7), 2618-2629.

Ulaby, F. T., and D. G. Long (2014), *Microwave radar and radiometric remote sensing*, 984 pp., The University of Michigan Press, Ann Arbor.

Vittucci, C., G. Vaglio Laurin, G. Tramontana, P. Ferrazzoli, L. Guerriero, and D. Papale (2019), Vegetation optical depth at L-band and above ground biomass in the tropical range: Evaluating their relationships at continental and regional scales, *International Journal of Applied Earth Observation and Geoinformation*, *77*, 151-161.

---

## Referee Report (RR1)

I want to thank the authors for their diligent work and responses to my comments. While the paper's clarity has been improved concerning the previous version, I felt I needed to clarify and reemphasize some of my earlier comments. In this round of the review, I provide my significant disagreements, and then I provide detailed comments on specific text in the manuscript. Also, I appreciate the authors' effort to make the assumptions 100% transparent throughout the manuscript, but this is not a favor; it is expected from every scientific manuscript.

I'm afraid I have to disagree with the major assumption, "*Forest attenuation due to leaves dominates at L-band*," as I stated in my previous review. The previous research overwhelmingly showed that "branches are the most dominant constituent for L-band attenuation and scattering of a forest canopy (please see Ferrazzoli and Guerriero, 1996, Ferrazzoli et al., 2002, Kurum et al., 2009)." I am adding a few excerpts from these papers:

Kurum et al, 2019 (DOI: 10.1109/TGRS.2009.2026641)

"*The contribution of leaves at L-band to the total backscattering response was found to be negligible compared with the contribution from branches and trunks. The leaves, however, were included in the simulations due to their significant effect on the canopy extinction.*"

"*Branches are main contributor to the volume scattering.*"

"*The effect of the trunk layer in the total extinction at L-band is relatively small due to the large densities of branches in the crown layer and the small heights of tree trunks [11]. This is typically true for two-layer forest canopies where the trunk layer is composed of vertical trunks. For the deciduous trees considered in this paper, the attenuation in the trunk layer was found about 1 dB. This trunk attenuation factor is assumed to be negligible.*"

Ferrazzoli et al., 2002 (DOI: 10.1109/TGRS.2002.807577)

"*Figs. 5 and 6 report, respectively, the model-simulated emissivities and transmissivities of trunks, branches, needles, and understory (in Fig. 5 soil contribution is also shown). The figures show that from the various forest components, branches mainly contribute to both emissivity and attenuation, while needles, trunks, and understory are minor contributors.We note that trunks, although containing most biomass, produce little effects; on the opposite, branches, which represent only a small percentage (10% to 30%) of the total biomass, are the main elements responsible for wave extinction and emission.*"

"*A detailed analysis indicated that, at L-band, the main contribution to emission and attenuation was due to forest branches, while trunks had smaller effects.*"

P. Ferrazzoli; L. Guerriero 1996 (DOI: 10.1109/36.485121)

"*Model predicted vegetation attenuation is plotted in Fig. 6 as a function of frequency, in the range 1-10 GHz, for a deciduous forest with high biomass (240 tonsha), at B = 15" and B = 45". The total attenuation and the single contributions of leaves, branches and trunks are shown. Leaf attenuation is low at 1 GHz and appreciably increases with frequency. Branch attenuation, on the contrary, is slightly affected by frequency. These results are in good agreement with experimental data of [26], which indicate crown attenuation of a deciduous forest to increase with frequency in summer time (with leaves) and to be almost frequency independent in winter time (without leaves). According to Fig. 6, when the frequency*"

*increases from 1 to 10 GHz, variations in the overall attenuation are low in absence of leaves, while are higher but still limited (-5 dB at 15", -10 dB at 45") in presence of leaves, although a forest with high biomass has been considered."*

The authors provided further evidence for the claim of dominant leaf contribution by referring Matzler (1994), Seele-Dunne et al. (2012), and Schwank et al. (2021). However, Matzler (1994) mainly studied transmissivity at frequencies above 5 GHz. It is challenging to extrapolate his results to L-band. Seele-Dunne et al. (2012) looked at backscatter data at C-band and did some simulations for a specific tree (trembling aspen) at L-band, and hard to generalize for a global conclusion on leaves dominance on the VOD. Schwank et al. (2021) acknowledge the dominance of branches by stating, "*Branches are most determinative for L-VOD of a forest canopy (Ferrazzoli and Guerriero, 1996).*" However, they violate the validity of the Maxwell Garnet mixing rule by formulating a canopy layer mixing model. Maxwell Garnet mixing rule is only valid within the limits of the quasi-static approach (the scattering losses are not incorporated), which is true when the wavelength is larger than the dominant inclusions.

The authors completely ignore contributions from branches with the assumption of leaf dominance in forest attenuation. This also leads to the misintereptration of the whole canopy average VOD results. It is well known that most biomass is confined within trunks and branches. As clearly articulated in the paper, VOD has a consequence of aggregate due to vegetation water content (short-term fluctuations) and dry matter (seasonal variations). Thus, it is contradictory to attribute the leaf water content to the whole canopy's average VOD. In addition, a high correlation between VOD and leaf water content does not necessarily mean that the leaf water content dominates VOD. From the present analysis, it is unclear which part of the plant influences VOD more, as the study completely ignores contributions from branches.

To sum up, it is known that the branches are the dominant constituents in L-band attenuation and volume scattering. Hence any forest model that only considers leaves cannot accurately mimic the physics (wave interaction and propagation). Thus, I can't entirely agree with using the leaf dielectric-mixing model to analyze experimental data at L-band.

Specific comments:

Page 2, Line 46: Microwave remote sensing is generally characterized by active (radar), passive (radiometry), and signals of opportunity (e.g., GNSS-R). GNSS-R has been mentioned on line 132 on page 5. You may want to move these up where you describe microwave remote sensing on page 2.

Page 3, Lines 77-82: Authors correctly argue that VOD is a better proxy for forest height and biomass than optical indexes by citing literature. Then, they compare their VOD results against the enhanced vegetation index derived from 30-meter satellite images of Sentinel 2 in section 4.1 and page 32, Line 623. A clarification would be helpful.

Page 6, Lines 135-136. This is not necessarily correct. The GNSS signals under dense forests can be smaller than GNSS reflections from wet surfaces.

Page 7, Line 159. Please define the acronym RHCP.

Page 9, Lines 203-204. It is also important to point out under what conditions these statements are correct. Otherwise, it sounds like these are general conclusions. As I mentioned in the previous review, it is hard to generalize Guerriero et al. (2020) result as it is only done for a particular case.

Page 10, Line 234: RHCP backscatter?

Page 11, Line 252: Is it SNR or CN0? https://insidegnss.com/measuring-gnss-signal-strength/

Page 14, Lines 296-298: This is a very strong statement regarding the dominance of direct signals. This is a very significant assumption for this paper and needs to be clearly backed. I suggest that the authors point out the similarities and differences between Guerriero et al. (2020) and the present setups. It is too fast to state that both setups are similar. In addition, geodetic-grade antennas are designed to reject reflected multipath coming from the lower hemisphere. The volume scattering is also a multipath and is received through the antenna's upper hemisphere. Yes, a significantly reduced ground-reflected multipath can be considered of second order, but a multipath due to the volume scattering can still be significant. Without a piece of concrete evidence on volume scattering, the interpretation of the results will be speculative.

Page 15, Lines 315-319: I would not call the negative difference unphysical; as you correctly stated, individual measurements include (1) random noise (slight system, and configuration differences) and (2) multipath interference. The negative difference mostly happens when the transmitter is visible through the gaps in the canopy. The supporting figure 1 (in the response letter) only shows the random noise in an open field; this does not necessarily match those happening within the gaps. Then, I can easily argue that other noise components (the multipath interference) might be due to the volume scattering since it provides an additional signal for under-canopy measurements through the gaps.

Page 23, Lines 449-450. It is hard to generalize any results with a few leaf samples. Figure 9(b) shows only three points on two different days.

Page 29, Line 552 - Page 30, Line 563: This is my main disagreement with this paper, as I stated in my general comment in the beginning.

Page 30, Figure 11(a): Again, it is hard to arrive at any conclusion with three medians of leaf measurements.

Page 31, Line 593: 10.9 kg m^-2 is an excessively high value since the mixing model only considers leaves. The cited values in the literature are the AGB of the whole canopy (trunks, branches, and leaves).

Page 31, Line 596. I disagree that "L-band VOD is primarily sensitive to leaves," as I stated in the beginning.

Page 36, Lines 691-693. Wave interactions at X-band (~3cm) and L-band (~20 cm) are fundamentally different. I would not compare the results at these two distinct frequencies.

---

## Author Response (AR2)

**Contents of the response document**

We reproduce each comment, followed by our response and the corresponding manuscript changes.

Line numbers in comments correspond to the original preprint. Line numbers in our responses correspond to the *revised* manuscript.

**1. Response to comments by Referee #1**

I am satisfied with the authors' response to my comments. I recommend that the paper is accepted for publication.

One very minor suggestion: I would recommend including the explanation provided in the response to the following comment in the final manuscript.

Reviewer comment: Lines 495 to 507: Why is it necessary to optimize v_veg with a daily time step? In a forest, in particular, this quantity is likely to vary over much longer time scales. This could obviate the need for some of the low-pass filtering in later steps.

Author response: That's a good suggestion, in fact, calibrating v_veg at a longer time scale is what we did initially. However, optimizing v_veg at a daily time step provides a more efficient way of mitigating the influence of outliers (like the rainfall event at the beginning of the time series). With a daily estimate, only the v_veg of a single day is heavily biased and this is easily removed with the low-pass filter. If we were to optimize v_veg over a moving period of say, a whole week, the whole week may be biased high around that event.

We thank the referee for the careful evaluation. We have included the following explanation:

*L541: Note that although calibrating v_veg over a time period longer than a day would also smooth the estimate, we found that optimizing v_veg at a daily time step and then applying a low-pass filter was much more effective in mitigating the influence of outliers.*

**2. Response to comments by Referee #2**

I want to thank the authors for their diligent work and responses to my comments. While the paper's clarity has been improved concerning the previous version, I felt I needed to clarify and reemphasize some of my earlier comments. In this round of the review, I provide my significant disagreements, and then I provide detailed comments on specific text in the manuscript. Also, I appreciate the authors' effort to make the

assumptions 100% transparent throughout the manuscript, but this is not a favor; it is expected from every scientific manuscript.

We thank the referee for the careful evaluation and detailed comments.

I'm afraid I have to disagree with the major assumption, "Forest attenuation due to leaves dominates at L-band," as I stated in my previous review. The previous research overwhelmingly showed that "branches are the most dominant constituent for L-band attenuation and scattering of a forest canopy (please see Ferrazzoli and Guerriero, 1996, Ferrazzoli et al., 2002, Kurum et al., 2009)." I am adding a few excerpts from these papers:

We acknowledge your disagreement with the statement "Forest attenuation due to leaves dominates at L-band". We note that this statement does not appear anywhere in our manuscript. Our manuscript does not state that forest attenuation due to leaves dominates at L-band and is in fact much more nuanced, see e.g.:

L198: *"Note that this formulation and the overall concept of bulk coefficients is applicable only when the inclusions in the canopy (i.e. the pockets of water within the vegetation tissues) are smaller in size compared to the observation wavelength (here $\lambda_0 \approx 19 \, cm$ for the GPS L1 frequency), so that scattering effects are small enough to be neglected [also see Jackson and Schmugge, 1991; Ulaby and Long, 2014]. While this may be true for leaves, this assumption may not hold for larger elements such as branches and trunks."*

L232: *"It is important to note that this simple formulation neglects several aspects, including volume scattering, which may be important in configurations with denser biomass (and also when interpreting LHCP backscatter or LHCP signals)."*

L554: *"Before we interpret these results, some limitations to the presented approach need to be emphasized. First, the dielectric model of Ulaby and El-rayes (1987) was originally developed for leaves, however, it is clear that branches and stems may also contribute to canopy extinction."*

L563: *"It is assumed that the dielectric model of leaves and its sensitivity to moisture, temperature, and salinity provides a sufficient approximation for the behaviour of the whole crown (branches and stems included)."*

Thus our actual assumption is that a model appropriate for a canopy constituted of small elements provides a reasonable first-order approximation, even though large branches, and to a lesser extent, trunks, also contribute to attenuation by the canopy in some proportions, the determination of which lies outside the scope of this study (and remains an active area of research in our opinion).

We discuss the implications of this assumption at several points in the manuscript and recognize that using other, more detailed, retrieval approaches may be helpful in the future, should the data be able to support the added complexity and higher degrees of freedom.

Kurum et al, 2019 (DOI: 10.1109/TGRS.2009.2026641)

"The contribution of leaves at L-band to the total backscattering response was found to be negligible compared with the contribution from branches and trunks. The leaves, however, were included in the simulations due to their significant effect on the canopy extinction."

We note that the beginning of this quote refers to backscatter and not attenuation of the direct signal as seen from the perspective of a sub-canopy receiver. The significant contribution of branches to backscatter is already acknowledged twice in our manuscript, at L232 and L558. The second part of the quote is in line with our assumption.

"Branches are main contributor to the volume scattering."

"The effect of the trunk layer in the total extinction at L-band is relatively small due to the large densities of branches in the crown layer and the small heights of tree trunks [11]. This is typically true for two-layer forest canopies where the trunk layer is composed of vertical trunks. For the deciduous trees considered in this paper, the attenuation in the trunk layer was found about 1 dB. This trunk attenuation factor is assumed to be negligible."

The implication is mentioned explicitly at L599: *"Note that this estimate should be interpreted with the awareness that VOD-based estimates of AGB likely do not weigh all canopy constituents evenly."*

We note that these limitations similarly apply to a large body of published studies which have used VOD observations as a proxy to investigate forest dynamics and inter-annual variations in aboveground biomass stocks.

We also mention this at L613: *"As for AGB, it's important to keep in mind that the CWC estimate does not weigh all canopy constituents evenly. Comparisons with other studies are quite difficult here because the relationship between VOD and vegetation water content is poorly known for forests"*

Finally we now add the following statement immediately afterwards at L614: *"In particular, our retrieval assumes that the attenuation is dominated by small canopy elements, even though the contribution of large elements (like large branches or trunks) to CWC is likely not negligible."*

Ferrazzoli et al., 2002 (DOI: 10.1109/TGRS.2002.807577)

"Figs. 5 and 6 report, respectively, the model-simulated emissivities and transmissivities of trunks, branches, needles, and understory (in Fig. 5 soil contribution is also shown). The figures show that from the various forest components, branches mainly contribute to both emissivity and attenuation, while needles, trunks, and understory are minor contributors. We note that trunks, although containing most biomass, produce little effects; on the opposite, branches, which represent only a small percentage (10% to 30%) of the total biomass, are the main elements responsible for wave extinction and emission."

"A detailed analysis indicated that, at L-band, the main contribution to emission and attenuation was due to forest branches, while trunks had smaller effects."

> We note that this study is focusing on a maritime pine forest and discusses the case of a spaceborne radiometer. Our setup is in a broadleaf forest and considers right polarized GNSS signals measured below the canopy, a situation which we believe is closer to the study by Guerriero et al. (2020) or Steele-Dunne et al. (2012).

P. Ferrazzoli; L. Guerriero 1996 (DOI: 10.1109/36.485121)

"Model predicted vegetation attenuation is plotted in Fig. 6 as a function of frequency, in the range 1-10 GHz, for a deciduous forest with high biomass (240 tonsha), at B = 15" and B = 45". The total attenuation and the single contributions of leaves, branches and trunks are shown. Leaf attenuation is low at 1 GHz and appreciably increases with frequency. Branch attenuation, on the contrary, is slightly affected by frequency. These results are in good agreement with experimental data of [26], which indicate crown attenuation of a deciduous forest to increase with frequency in summer time (with leaves) and to be almost frequency independent in winter time (without leaves). According to Fig. 6, when the frequency increases from 1 to 10 GHz, variations in the overall attenuation are low in absence of leaves, while are higher but still limited (-5 dB at 15", -10 dB at 45") in presence of leaves, although a forest with high biomass has been considered."

> Thank you for pointing us to this reference. We acknowledge that the conclusions of this model-based study are different compared to the (also model-based) conclusions of Steele-Dunne et al. 2012. We add the following statement (underlined) to L561:
>
> *"Steele-Dunne et al. (2012) arrived at similar conclusions and concluded that leaf moisture is by far the dominant control on vegetation transmissivity at L-band for both polarizations (but see Ferrazzoli and Guerriero (1996) for a different perspective)."*

The authors provided further evidence for the claim of dominant leaf contribution by referring Matzler (1994), Seele-Dunne et al. (2012), and Schwank et al. (2021).

However, Matzler (1994) mainly studied transmissivity at frequencies above 5 GHz. It is challenging to extrapolate his results to L-band. Seele-Dunne et al. (2012) looked at backscatter data at C-band and did some simulations for a specific tree (trembling aspen) at L-band, and hard to generalize for a global conclusion on leaves dominance on the VOD. Schwank et al. (2021) acknowledge the dominance of branches by stating, "Branches are most determinative for L-VOD of a forest canopy (Ferrazzoli and Guerriero, 1996)." However, they violate the validity of the Maxwell Garnet mixing rule by formulating a canopy layer mixing model. Maxwell Garnet mixing rule is only valid within the limits of the quasi-static approach (the scattering losses are not incorporated), which is true when the wavelength is larger than the dominant inclusions.

We would like to emphasize that nowhere in the manuscript do we *claim* that leaves have a dominant contribution to attenuation. Instead, our assumption is that a model appropriate for a canopy constituted of small elements (such as leaves) provides a reasonable first-order approximation. See e.g.:

L198: *"Note that this formulation and the overall concept of bulk coefficients is applicable only when the inclusions in the canopy (i.e. the pockets of water within the vegetation tissues) are smaller in size compared to the observation wavelength (here $\lambda_0 \approx 19$ cm for the GPS L1 frequency), so that scattering effects are small enough to be neglected [also see Jackson and Schmugge, 1991; Ulaby and Long, 2014]. While this may be true for leaves, this assumption may not hold for larger elements such as branches and trunks."*

L563: *"It is assumed that the dielectric model of leaves and its sensitivity to moisture, temperature, and salinity provides a sufficient approximation for the behaviour of the whole crown (branches and stems included)."*

The authors completely ignore contributions from branches with the assumption of leaf dominance in forest attenuation. This also leads to the misintereptration of the whole canopy average VOD results. It is well known that most biomass is confined within trunks and branches. As clearly articulated in the paper, VOD has a consequence of aggregate due to vegetation water content (short-term fluctuations) and dry matter (seasonal variations). Thus, it is contradictory to attribute the leaf water content to the whole canopy's average VOD. In addition, a high correlation between VOD and leaf water content does not necessarily mean that the leaf water content dominates VOD. From the present analysis, it is unclear which part of the plant influences VOD more, as the study completely ignores contributions from branches.

We note that none of our findings or conclusions implies (or aims to make) any statement on which part of the plant influences VOD the most and how to best deal with it. We believe this remains an area of active research which will hopefully benefit from more widespread and continuous in situ measurements of in situ VOD.

To sum up, it is known that the branches are the dominant constituents in L-band attenuation and volume scattering. Hence any forest model that only considers leaves cannot accurately mimic the physics (wave interaction and propagation). Thus, I can't entirely agree with using the leaf dielectric-mixing model to analyze experimental data at L-band.

Specific comments:

Page 2, Line 46: Microwave remote sensing is generally characterized by active (radar), passive (radiometry), and signals of opportunity (e.g., GNSS-R). GNSS-R has been mentioned on line 132 on page 5. You may want to move these up where you describe microwave remote sensing on page 2.

> Thank you for this suggestion. As page 2 aims to provide a broad introduction, we prefer to discuss GNSS-R after we have introduced global navigation satellite systems in general on page 5.

Page 3, Lines 77-82: Authors correctly argue that VOD is a better proxy for forest height and biomass than optical indexes by citing literature. Then, they compare their VOD results against the enhanced vegetation index derived from 30-meter satellite images of Sentinel 2 in section 4.1 and page 32, Line 623. A clarification would be helpful.

> Optical vegetation indices still provide useful information on vegetation development and are not used here as validation for VOD. This is why differences between VOD and EVI are discussed in section 4.1:
>
> *L403: "EVI is a commonly used vegetation index and an overall proxy for vegetation greenness, health, and photosynthetic activity. Generally, we find that the temporal evolution of VOD appears to lag behind that of EVI by about 2 months. This is consistent with previous findings over drylands by Tian et al. (2016) who found a temporal shift (increasing as a function of forest density) between satellite-based VOD and vegetation greenness."*
>
> The discussion on page 32, line 623 refers to NDWI and not EVI. Here, NDWI is used as an independent data source, not as a proxy for height or biomass:
>
> *L630: "NDWI is a good proxy for vegetation water content and is based on optical and near-infrared measurements, thus providing fully independent observations with respect to our retrieval."*

Page 6, Lines 135-136. This is not necessarily correct. The GNSS signals under dense forests can be smaller than GNSS reflections from wet surfaces.

Thanks for noting this, we have made our statement more accurate (underlined):

L135: *"GNSS reflectometry relies on GNSS signals that are reflected from the Earth's surface and which are weaker than the open-sky GNSS signals used as reference here."*

Page 7, Line 159. Please define the acronym RHCP.

Thanks, we moved the acronym's first definition to that line.

Page 9, Lines 203-204. It is also important to point out under what conditions these statements are correct. Otherwise, it sounds like these are general conclusions. As I mentioned in the previous review, it is hard to generalize Guerriero et al. (2020) result as it is only done for a particular case.

We modify the statement as follows (underlined):

L203: *"However, using a more complex theoretical scattering model, Guerriero et al. (2020) showed (for the case of a poplar forest) that the RHCP GNSS signals measured below a forest canopy are dominated by coherent attenuation whereas only the left-hand circular polarized signals (which most geodetic ground-based GNSS antennas are designed to reject) are dominated by volume scattering."*

Page 10, Line 234: RHCP backscatter?

Thanks a lot for spotting this, changed to LHCP.

Page 11, Line 252: Is it SNR or CN0? https://insidegnss.com/measuring-gnss-signal-strength/

The receivers log $C/N_0$. For simplicity we used the notation 'SNR' as it does not introduce a fraction in the equations and is the most common notation (e.g. the same is done in https://doi.org/10.1007/s10291-012-0259-7 or https://doi.org/10.1038/s41598-019-40456-2). But it's a good point to mention this explicitly, thanks.

L254: *"The quantity logged by the Septentrio receiver is the carrier-to-noise density ratio ($C/N_0$), which we report as SNR for simplicity, assuming a 1-Hz bandwidth [Larson and Nievinski, 2012]."*

Page 14, Lines 296-298: This is a very strong statement regarding the dominance of direct signals. This is a very significant assumption for this paper and needs to be clearly backed. I suggest that the authors point out the similarities and differences between Guerriero et al. (2020) and the present setups. It is too fast to state that both setups are similar. In addition, geodetic-grade antennas are designed to reject reflected

multipath coming from the lower hemisphere. The volume scattering is also a multipath and is received through the antenna's upper hemisphere. Yes, a significantly reduced ground-reflected multipath can be considered of second order, but a multipath due to the volume scattering can still be significant. Without a piece of concrete evidence on volume scattering, the interpretation of the results will be speculative.

Maybe this was a misunderstanding. We modify the statement (underlined) to make it clear this refers to ground multipath (which is discussed earlier):

*L298: "Thus, in our case, the difference in SNR between the two sites is predominantly due to the attenuation of the direct RHCP signal by the forest canopy and it is reasonable to assume that ground multipath effects are of second order"*

Page 15, Lines 315-319: I would not call the negative difference unphysical; as you correctly stated, individual measurements include (1) random noise (slight system, and configuration differences) and (2) multipath interference. The negative difference mostly happens when the transmitter is visible through the gaps in the canopy. The supporting figure 1 (in the response letter) only shows the random noise in an open field; this does not necessarily match those happening within the gaps. Then, I can easily argue that other noise components (the multipath interference) might be due to the volume scattering since it provides an additional signal for under-canopy measurements through the gaps.

We update the statement as follows:

*L317: "In some cases, the instantaneous transmissivity values computed from the raw GNSS measurements were higher than 1, leading to a VOD lower than zero (about 8% of all measurements).*

Page 23, Lines 449-450. It is hard to generalize any results with a few leaf samples. Figure 9(b) shows only three points on two different days.

We do not believe our statement is implying any generalization.

*L451: "The leaf samples collected on the site in October also confirm that some intra-day variability exists in relative leaf water content (Fig. 9b)."*

Besides, diurnal variability in leaf water content has been widely reported (see the studies cited at this point in the text).

Page 29, Line 552 - Page 30, Line 563: This is my main disagreement with this paper, as I stated in my general comment in the beginning.

As mentioned our general response above, we modify one sentence in this section as follows:

*L559: "Steele-Dunne et al. (2012) arrived at similar conclusions and concluded that leaf moisture is by far the dominant control on vegetation transmissivity at L-band for both polarizations, (but see Ferrazzoli and Guerriero (1996) for a different perspective)."*

We also modify L554 as follows:

*L554: "Before we interpret these results, some limitations to the presented approach need to be emphasized. First, the dielectric model of Ulaby and El-rayes (1987) was originally developed for leaves, however, it is clear that branches and stems  also contribute to canopy extinction."*

Page 30, Figure 11(a): Again, it is hard to arrive at any conclusion with three medians of leaf measurements.

See our statement a few lines later:

*L574: "this does not provide any formal validation of the $m_g$ retrieval"*

Page 31, Line 593: 10.9 kg m^-2 is an excessively high value since the mixing model only considers leaves. The cited values in the literature are the AGB of the whole canopy (trunks, branches, and leaves).

The retrieved AGB is not that of leaves only since the VOD measurements also include the contribution from larger elements like branches as mentioned in the manuscript and as argued earlier by the referee. It is only that the retrieved AGB is estimated with a simple model which is physically valid for electrically small canopy constituents and not necessarily fully appropriate for the larger elements (and this latter part is the main assumption).

The cited values are fully comparable since they stem from empirical relationships derived between L-band VOD and biomass reference measurements, and similarly do not individually resolve the uneven contributions of leaves, branches and trunks to VOD:

*"L605: For instance, using the exponential relationship calibrated at L-band by Vittucci et al. [2019], we obtain (for an average VOD value of 0.79 at our site) an AGB of 13.8 kg m-2, not so far from our estimate."*

Page 31, Line 596. I disagree that "L-band VOD is primarily sensitive to leaves," as I stated in the beginning.

This sentence (L601) reports the results of Steele-Dunne et al. 2012. We remove *"primarily"*.

Page 36, Lines 691-693. Wave interactions at X-band (~3cm) and L-band (~20 cm) are fundamentally different. I would not compare the results at these two distinct frequencies.

We reformulate as:

*L698: "These results indicate that L-band VOD is quite sensitive to intercepted rainfall, as has been shown with X-band VOD observations from the AMSR-E satellite (Xu et al., 2021) and from an in-situ radiometer (Schneebeli et al., 2011)."*